# TRPA1 activation in non-sensory supporting cells contributes to regulation of cochlear sensitivity after acoustic trauma

A. Catalina Vélez-Ortega ®[1] ✉, Ruben Stepanyan[1,3], Stephanie E. Edelmann[1], Sara Torres-Gallego[1], Channy Park[2], Desislava A. Marinkova[1,4], Joshua S. Nowacki[1], Ghanshyam P. Sinha ®[1,5] & Gregory I. Frolenkov ®[1] ✉

TRPA1 channels are expressed in nociceptive neurons, where they detect noxious stimuli, and in the mammalian cochlea, where their function is unknown. Here we show that TRPA1 activation in the supporting non-sensory Hensen's cells of the mouse cochlea causes prolonged $Ca^{2+}$ responses, which propagate across the organ of Corti and cause long-lasting contractions of pillar and Deiters' cells. Caged $Ca^{2+}$ experiments demonstrated that, similar to Deiters' cells, pillar cells also possess $Ca^{2+}$-dependent contractile machinery. TRPA1 channels are activated by endogenous products of oxidative stress and extracellular ATP. Since both these stimuli are present in vivo after acoustic trauma, TRPA1 activation after noise may affect cochlear sensitivity through supporting cell contractions. Consistently, TRPA1 deficiency results in larger but less prolonged noise-induced temporary shift of hearing thresholds, accompanied by permanent changes of latency of the auditory brainstem responses. We conclude that TRPA1 contributes to the regulation of cochlear sensitivity after acoustic trauma.

The transient receptor potential cation channel A1 (TRPA1) is expressed in somatosensory nociceptive neurons[1,2], where it can be activated by various pungent compounds[3–5]. TRPA1 is essential for nociceptive responses and inflammatory pain in vivo[6,7]. Inflammatory agents such as bradykinin can activate TRPA1 channels indirectly through activation of phospholipase C[4]. In addition, TRPA1 channels may sense a variety of cellular stresses due to their gating by intracellular $Ca^{2+}$ [8–10] or by endogenous products of oxidative stress such as the lipid peroxidation byproduct, 4-hydroxynonenal (4-HNE)[11,12]. Thus, neuronal TRPA1 channels seem to be universal "damage sensors" that integrate various signals during tissue damage (reviewed in ref. 13).

TRPA1 is also present in the cochlea[14–17] and it was originally proposed to be the mechanotransducer channel of the mammalian hair cells[2,15]. However, mice lacking functional TRPA1 channels ($Trpa1^{-/-}$) exhibit deficiencies in pain sensitivity but normal hearing thresholds, vestibular function, and hair cell mechanosensitivity[6,7]. Therefore, the function of TRPA1 channels in the inner ear remains elusive.

Here we show that, in addition to the sensory hair cells[16], functional TRPA1 channels are also present in many non-sensory (supporting) cells of the organ of Corti. Among them, Hensen's cells are the most sensitive to the direct application of various TRPA1 agonists to the endolymphatic side of the cochlear epithelium in both young postnatal and adult mice. TRPA1-initated $Ca^{2+}$ responses in Hensen's cells are long-lasting and propagate across the rows of sensory hair cells, causing shape changes in Deiters' and pillar cells. Direct increase of intracellular $Ca^{2+}$ with flash-photolysis evokes contractile responses

[1]Department of Physiology, College of Medicine, University of Kentucky, Lexington, KY 40536, USA. [2]Department of Head & Neck Surgery, David Geffen School of Medicine, UCLA, Los Angeles, CA 90095, USA. [3]Present address: Department of Otolaryngology, Case Western Reserve University, Cleveland, OH 44106, USA. [4]Present address: Department of Pharmacology and Toxicology, University of Arkansas for Medical Sciences, Little Rock, AR 72205, USA. [5]Present address: Department of Anesthesiology and Perioperative Medicine, University of Pittsburgh, Pittsburgh, PA 15261, USA. ✉e-mail: catavelezo@uky.edu; Gregory.Frolenkov@uky.edu

in Deiters' and pillar cells. In vivo, TRPA1-initiated contraction of supporting cells would cause changes of the geometry and/or stiffness of the organ of Corti, increasing the hearing thresholds. These effects are expected to be long-lasting since endogenous TRPA1 agonists are generated after acoustic trauma for several days. Consistently, we found that moderate noise exposure causes larger but less prolonged temporary shifts of hearing thresholds (TTS) in $Trpa1^{-/-}$ mice compared to their wild-type littermates. In addition, $Trpa1^{-/-}$ mice exhibit changes of the latency of the supra-threshold auditory brainstem responses (ABRs) even after complete recovery of hearing thresholds. Thus, our data establish TRPA1 channels in the cochlea as "damage sensors" initiating protective cochlear responses contributing to noise-induced TTS.

## Results

### TRPA1 channels are widely expressed in the cochlear sensory epithelium

The cochlear sensory epithelium contains mechanosensory inner and outer hair cells (IHCs and OHCs, respectively) and several types of supporting cells (Fig. 1a). $Trpa1$ mRNA has been detected in both hair cells and supporting cells[14,15]. Unfortunately, all TRPA1 antibodies that we tested exhibited non-specific staining in the cochleae of the previously generated TRPA1-deficient ($Trpa1^{-/-}$) mice[7]. Therefore, we explored the expression of the human placental alkaline phosphatase (PLAP) reporter that was placed under control of the endogenous $Trpa1$ promoter in the $Trpa1^{-/-}$ mice[7]. PLAP expression was detected in IHCs and OHCs as well as in most types of supporting cells, including cells of the Kolliker's organ and pillar, Deiters', Hensen's, and Claudius' cells (Fig. 1b, c). This PLAP labeling can be interpreted as an indication for potential TRPA1 expression, but it does not demonstrate the expression of functional TRPA1 channels. Likewise, it cannot provide any information on the intracellular sorting of TRPA1 channels between apical and basal sides of the sensory epithelium, which are separated by tight junctions. Therefore, more direct functional assessments are needed.

### Hensen's cells are the major "sensors" of TRPA1 agonists in the organ of Corti

With the exception of OHCs[16], it is still unclear which cells of the cochlea possess functional TRPA1 channels. We used ratiometric $Ca^{2+}$ imaging to explore cellular responses evoked by local delivery of the endogenous TRPA1 agonist, 4-HNE, to the endolymphatic (apical) side of the cochlear epithelium. Application of 4-HNE via a puff pipette produced robust $Ca^{2+}$ responses in Hensen's cells of young postnatal wild-type mice (Fig. 2a, top; Supplementary Video 1). These responses had three phases: (i) an initial small increase of cytosolic $Ca^{2+}$ concentration ($[Ca^{2+}]_i$), (ii) a subsequent prominent rise of $[Ca^{2+}]_i$ with an average delay of $38.1 \pm 2.0$ s from puff onset (184 cells, 7 explants) (Fig. 2b), and (iii) a slow decay after the response peak ($0.53 \pm 0.05\%$ per second, 161 cells, 7 explants). The decay was an order of magnitude slower than the speed of drug washout from the application site (~7.6% per second). Similar robust and long-lasting $Ca^{2+}$ responses were evoked in young postnatal wild-type Hensen's cells by cinnamaldehyde (CA) (Fig. 2d, top; Fig. 2e) and mustard oil (allyl isothiocyanate, AITC) (Supplementary Fig. 2a, b), which are selective but not endogenous TRPA1 agonists[4]. We also tested the responses of the cochlear epithelium to the potent TRPA1 agonist para-benzoquinone (pBQN)[18] and observed a robust increase in $[Ca^{2+}]_i$ in young postnatal wild-type Hensen's cells with a similar delay from puff onset ($38.5 \pm 3.7$ s, 2 explants) (Supplementary Fig. 2c, d). As expected, all TRPA1 agonists failed to induce any $Ca^{2+}$ responses in the young postnatal cochlear epithelium of $Trpa1^{-/-}$ mice, confirming the crucial role of TRPA1 channels in these responses (Fig. 2a, d, bottom; Fig. 2b, c, e, f; Supplementary Fig. 2d). The TRPA1-dependent $Ca^{2+}$ responses in Hensen's cells were attenuated in the low-$Ca^{2+}$ extracellular medium

(Supplementary Fig. 2e), which is consistent with the notion that these responses require $Ca^{2+}$ influx through the TRPA1 channels at the plasma membrane. The variability of $Ca^{2+}$ responses to different TRPA1 agonists is not surprising, because these agonists may have different binding sites, gating, and desensitization kinetics on TRPA1[11,19–21].

The expression of functional TRPA1 channels in the cochlea is not limited to the young postnatal animals. In the adult wild-type mice, puff application of CA to the endolymphatic side of the sensory epithelium also produced prominent and long-lasting $Ca^{2+}$ responses in Hensen's cells, which were never observed in $Trpa1^{-/-}$ littermates (Fig. 2g–i). In contrast to the responses in the young cells, CA-evoked $Ca^{2+}$ responses in the adult cells often oscillate (Fig. 2h), which may reflect the dependence of these responses on the intracellular $Ca^{2+}$ (see below) and a potential postnatal maturation of $Ca^{2+}$ buffering mechanisms in Hensen's cells, similar to the ones observed in hair cells[22].

Despite ubiquitous PLAP reporter labeling within the cochlea (Fig. 1b, c), none of the TRPA1 agonists tested (4-HNE, CA, AITC, and pBQN) evoked detectable $Ca^{2+}$ responses in the sensory hair cells or in the Claudius' cells located next to the Hensen's cells in radial directions (Fig. 2a, d, g, top; Supplementary Fig. 2a, c, Video 1). Likewise, supporting cells located closer to the modiolus than Hensen's cells also did not directly respond to TRPA1 agonists, even though some of them, like the cells of the Kolliker's organ, exhibit spontaneous $Ca^{2+}$ oscillations in young cochlear epithelia (see[23] and Supplementary Fig. 1a, b). Deiters' cells exhibited delayed $Ca^{2+}$ responses only after repeated stimulation with TRPA1 agonists, likely indicating an indirect effect. Based on these data and on our previous study demonstrating the expression of functional TRPA1 channels on the basolateral (perilymphatic) but not on the apical (endolymphatic) plasma membrane of OHCs[16], we believe that Hensen's cells are the main cells that express TRPA1 channels at the endolymphatic side of the cochlear epithelium. Thus, Hensen's cells are the major sensors of the endogenous byproducts of oxidative damage, such as 4-HNE, in the scala media. Our data do not refute, however, the possibility that the TRPA1 channels in other cell types may sense these byproducts in the scala tympani.

### TRPA1-initiated $Ca^{2+}$ responses in Hensen's cells depend on intracellular $Ca^{2+}$

The long-lasting increase of $[Ca^{2+}]_i$ in the Hensen's cells after TRPA1 activation by 4-HNE or CA eventually declined and the cells responded to subsequent stimuli (Fig. 3a). Successive stimulations led to a larger or a smaller $Ca^{2+}$ response in the same cell depending on the $[Ca^{2+}]_i$ prior to the stimulation. The cell was usually more responsive to the TRPA1 agonist when its pre-stimulus $[Ca^{2+}]_i$ was in an intermediate range of ~140–220 nM (Fig. 3b). These results are consistent with the notion that TRPA1 channels are potentiated by a moderate increase of $[Ca^{2+}]_i$ and inactivated by very high $[Ca^{2+}]_i$[2,9,24]. After TRPA1 activation, cells often recovered to slightly higher $[Ca^{2+}]_i$ levels relative to the pre-stimulus baseline (Fig. 3a, compare $[Ca^{2+}]_i$ at 100 vs. 400 s). This elevated $[Ca^{2+}]_i$ may have potentiated the responses to the subsequent stimulations until the baseline $[Ca^{2+}]_i$ reached the threshold of TRPA1 inactivation by intracellular $Ca^{2+}$ (Fig. 3b). Consistently, responding cells with an intermediate pre-stimulus $[Ca^{2+}]_i$ exhibited shorter delays in $Ca^{2+}$ responses (Fig. 3b, inset).

### TRPA1 channels can also be activated by extracellular ATP

TRPA1 channels can be activated downstream of a variety of G-protein coupled receptors, including bradykinin and metabotropic acetylcholine receptors[3,4]. It is reasonable to speculate that TRPA1 can be also activated downstream of the G-protein coupled purinergic receptors (P2Y type) that are widely expressed in the cochlea and are thought to be activated after noise-induced trauma[25]. However, activation of TRPA1 channels by extracellular ATP has never been demonstrated. Therefore, we transfected human embryonic kidney (HEK) 293 cells, which express endogenous P2Y receptors[26], with a

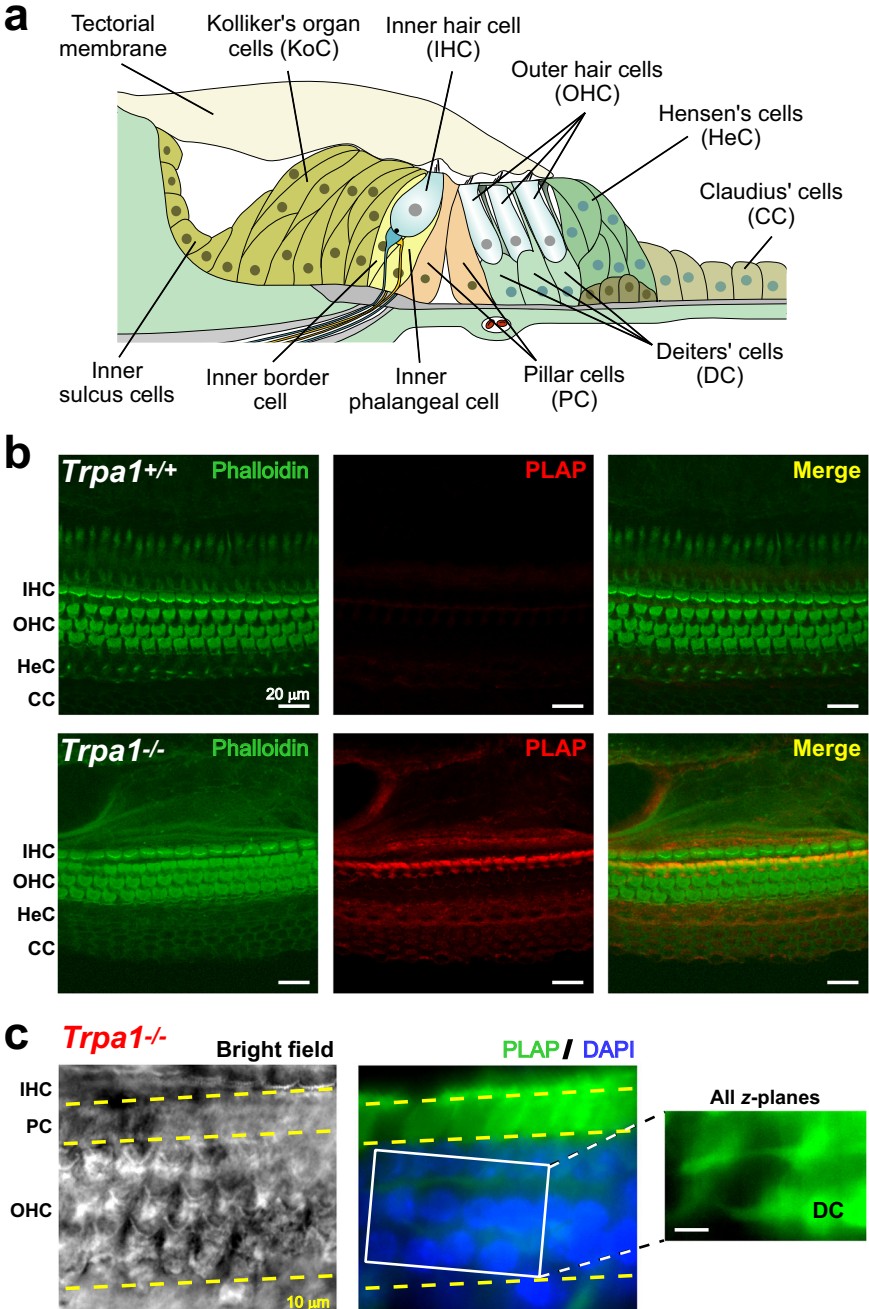

**Fig. 1 | Expression of TRPA1 in the cochlear epithelium. a** Diagram of the cochlear epithelium in young postnatal mice (<P7) illustrating mechanosensory inner (IHC) and outer (OHC) hair cells on top of inner and outer phalangeal cells, the latter known as Deiters' cells (DC). Next to the IHCs, toward the modiolus, are the inner border cells, cells of the Kolliker's organ (KoC), and inner sulcus cells. Inner and outer pillars cells (PC) are located between IHCs and OHCs. Hensen's (HeC) and Claudius (CC) cells are found toward the lateral side of the organ of Corti. In the adult cochlea, the Kolliker's organ cells are replaced by (or differentiate into) inner sulcus cells, while pillar cells become separated and create the fluid-filled tunnel of Corti. **b** Confocal images of cochlear epithelia (P4) at low magnification in wild-type (top) and *Trpa1⁻/⁻* (bottom) littermates showing PLAP immunolabeling (red) and counterstaining of F-actin with phalloidin (green). The images were kindly provided by Drs. Kevin Kwan and David Corey. **c** PLAP immunolabeling (green) in a *Trpa1⁻/⁻* cochlear explant at P7. Left panel is a reference bright-field image at the focal plane of hair cell stereocilia. Middle panel shows a fluorescent confocal image at OHC nuclei (DAPI counterstaining, blue) and strong expression of PLAP in pillar cells. A maximum intensity *z*-axis projection (right panel) revealed PLAP labeling in Deiters' cell bodies and in their phalangeal processes (DAPI signal was omitted for clarity). Images are representative of 3 independent series. Here and everywhere below, dashed lines indicate boundaries between different cell types. Scale bars: 20 μm (**b**) and 10 μm (**c**). Source data are provided as a Source Data file.

bicistronic construct expressing mouse *Trpa1* and *GFP* (Fig. 4a). Whole-cell patch-clamp recordings revealed large inward currents upon TRPA1 activation with AITC in GFP/TRPA1-transfected cells but not in the non-transfected or mock-transfected (GFP only) cells (Fig. 4b). These results confirmed the effectiveness of transfection and the lack of endogenous TRPA1 channels in HEK293 cells. Then, we applied 10 μM of ATP and observed large ATP-evoked currents in GFP/TRPA1-expressing cells but not in the mock-transfected or non-transfected cells (Fig. 4c). We conclude that TRPA1 may be indirectly activated by extracellular ATP, most likely via endogenous P2Y

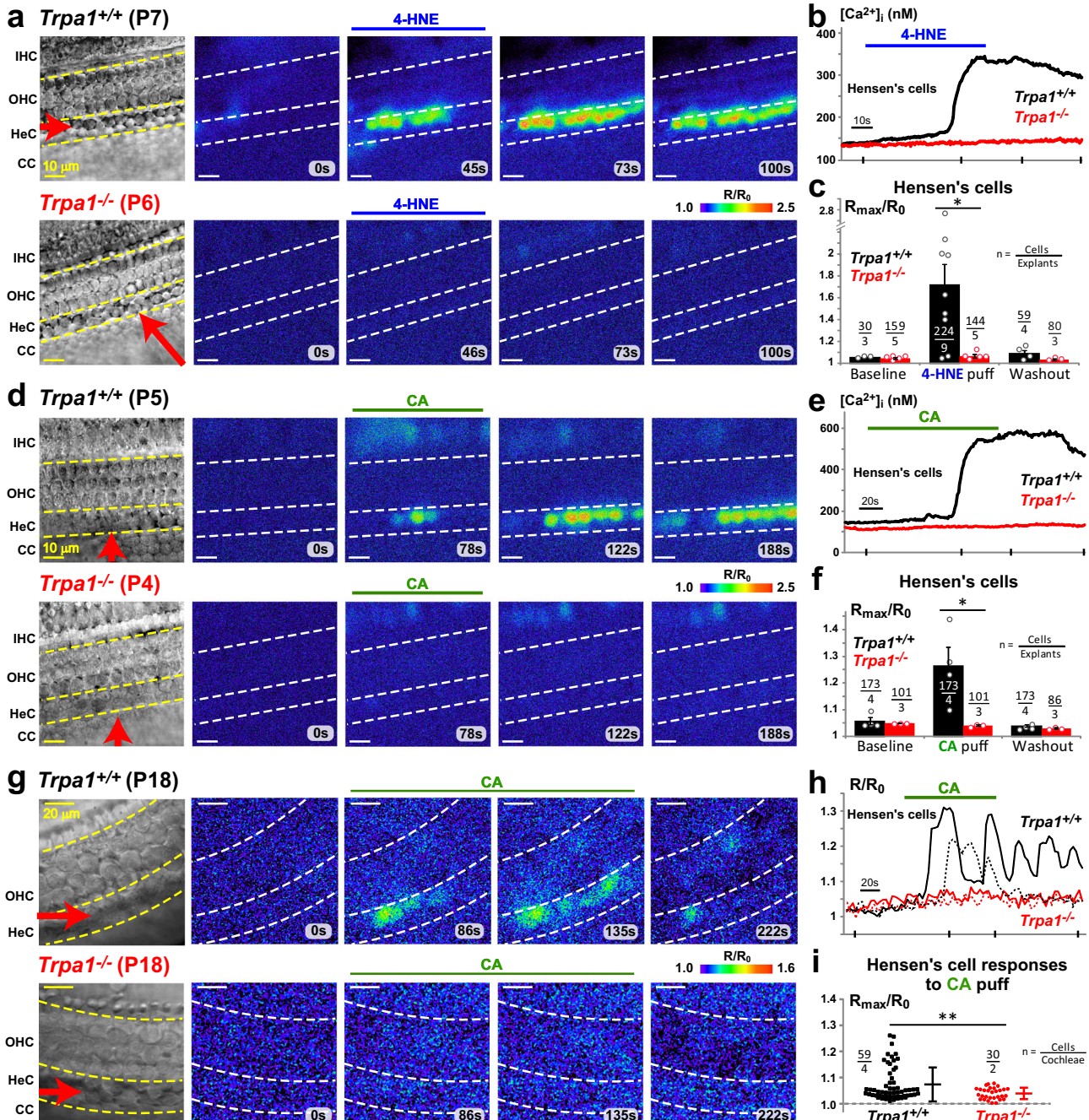

**Fig. 2 | Hensen's cells exhibit long-lasting Ca²⁺ responses to TRPA1 agonists.**
**a–c** Ca²⁺ responses to puff application of 200 μM of 4-hydroxynonenal (4-HNE).
**a** Time-lapse fura-2 ratio images (R = F₃₄₀/F₃₈₀) normalized to the baseline (R₀)
showing responses to 4-HNE application in wild-type (top) and *Trpa1⁻/⁻* (bottom)
postnatal cochlear explants. Left, reference bright field images of the same field of
view. Puff pipettes were positioned above the tissue (red arrows) and application
timing is shown by horizontal bars above the images. **b** Representative changes of
cytosolic Ca²⁺ concentration evoked by 4-HNE in wild-type (black) and *Trpa1⁻/⁻*
(red) Hensen's cells. Ticks on X-axis show timing of frames in (**a**). **c** Normalized
amplitude (Rmax/R₀) of Ca²⁺ responses to 4-HNE in wild-type (black) and *Trpa1⁻/⁻*
(red) Hensen's cells. Quantification of Ca²⁺ responses at consecutive time windows
(95 s) during *Baseline* (before agonist application), *Puff* (starting with TRPA1

agonist application), and *Washout* (~3 min after puff application). The data are
shown as Mean ± SE. The asterisk indicates statistical significance (*P* = 0.025, two-
sided Student's *t* test). **d–f** Ca²⁺ responses to puff application of 100 μM CA. Layout
of panels as in (**a–c**). **f** The asterisk indicates statistical significance (*P* = 0.044, two-
sided Student's *t* test). **g–i** Ca²⁺ responses in adult cochleae to puff application of
200 μM CA. Layout of panels (**g–i**) is similar to (**a–c**), except that representative
traces are shown from two cells (**h**) and the statistics panel (**i**) shows Mean ± SD
bars plus scattergrams of all measured cells due to the variability of responses.
Asterisks indicate statistical significance (*P* = 0.0005, two-sided Welch's *t* test).
Supplementary Fig. 1a–c shows F₃₄₀/F₃₈₀ frames from panels (**a**), (**d**) and (**g**) before
R/R₀ normalization. Source data are provided as a Source Data file.

receptors. Thus, following noise exposure in vivo, TRPA1 channels in
the cochlea may respond to both, the relatively fast release of ATP into
cochlear fluids[27] and the prolonged accumulation of endogenous
TRPA1 agonists such as 4-HNE due to oxidative stress[28].

## Hensen's cell responses to TRPA1 agonists and extracellular ATP are different

Ca²⁺ responses evoked by extracellular ATP in the supporting cells are
known to involve activation of P2Y receptors, generation of inositol

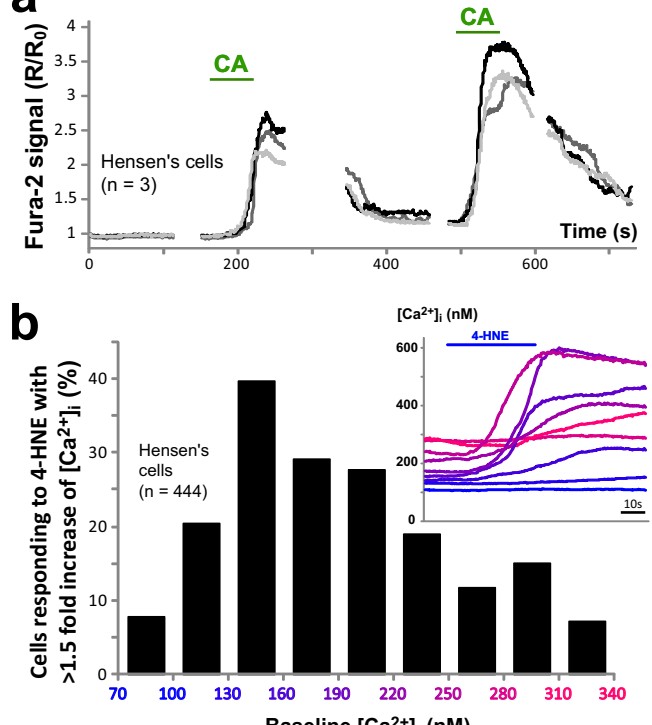

**Fig. 3 | Responses to TRPA1 agonists depend on intracellular Ca²⁺ levels.**
**a** Representative Ca²⁺ responses to two consecutive puff applications of CA (100 μM) in three wild-type Hensen's cells. The time scale is continuous throughout both applications, while the imaging was paused (breaks in the traces) to reduce photo-damage. **b** Relationship between percentage of cells responding to 400 μM of 4-HNE with at least a 1.5 fold increase of $[Ca^{2+}]_i$ (*y*-axis) and their pre-stimulus $[Ca^{2+}]_i$ (color-coded at *x*-axis). The inset shows Ca²⁺ response traces to several consecutive puff applications of 400 μM of 4-HNE in various Hensen's cells selected from one field of view. The traces are color-coded corresponding to the pre-stimulus levels of intracellular Ca²⁺. Notice that even the highest pre-stimulus $[Ca^{2+}]_i$ was still below the maximal $[Ca^{2+}]_i$ developed during TRPA1-initiated responses. Therefore, the lack of the responses to the TRPA1 agonist at high pre-stimulus $[Ca^{2+}]_i$ cannot be explained by saturation of the fura-2 dye. Source data are provided as a Source Data file.

triphosphate (IP₃), and Ca²⁺ release from intracellular stores (Supplementary Fig. 3a, left)[29–32]. In theory, these responses could be also modified by activation of Ca²⁺-permeable TRPA1 channels downstream of P2Y receptors (Fig. 4). However, we were not able to observe differences between ATP-evoked Ca²⁺ responses in supporting cells of wild-type and *Trpa1*⁻/⁻ mice, suggesting that TRPA1 contribution to ATP-evoked Ca²⁺ responses (if present) is minor in young postnatal mice. It is also worth mentioning that we applied extracellular ATP to the endolymphatic side of the cochlear epithelium, and we did not test the involvement of TRPA1 channels in modulation of responses evoked by ATP in the perilymph.

Ca²⁺ responses in Hensen's cells after application of extracellular ATP and TRPA1 agonists have very different dynamics (Supplementary Fig. 3b, c). Responses to ATP often oscillated and exhibited fast clearance (2.58 ± 0.8% per second; 29 cells, 2 explants) even at very high concentrations of ATP (*i.e.*, 250 μM). Such a fast decline of $[Ca^{2+}]_i$ was never observed after TRPA1 activation by 4-HNE, even when ATP and TRPA1 responses were compared in the same cell (Supplementary Fig. 3d, e). Thus, the slow clearance of $[Ca^{2+}]_i$ after TRPA1 agonists is not an intrinsic property of the Hensen's cells but probably the result of sustained TRPA1 activation (Supplementary Fig. 3a, right). This sustained activation may be caused by agonist-induced covalent

modifications of TRPA1[11,19,20] or by a positive "self-locking" feedback, in which Ca²⁺ ions entering the cell through TRPA1 channels keep open or re-activate these channels[8,9].

## TRPA1-initiated responses in Hensen's cells propagate across the organ of Corti and trigger Ca²⁺ waves in the Kolliker's organ

Despite spontaneous Ca²⁺ waves occurring in the proximal side of the cochlear epithelium after ~P4, we noticed that the responses of Hensen's cells to 4-HNE were often followed by Ca²⁺ waves in the Kolliker's organ. Therefore, we explored this phenomenon in neonatal (P1-P2) wild-type mice, which exhibit very few spontaneous Ca²⁺ waves in the Kolliker's organ[23]. As expected, application of 4-HNE to the Hensen's cells in these neonatal mice evoked prominent Ca²⁺ responses propagating along the length of the organ of Corti. However, these Ca²⁺ responses often spread across the organ of Corti and caused a fast-propagating Ca²⁺ wave in the Kolliker's organ (Fig. 5a, top). In contrast, application of 4-HNE to the Kolliker's organ in the same explant produced no response, even after several sequential applications (Fig. 5a, bottom). Thus, cells of the Kolliker's organ themselves do not respond to the TRPA1 agonist. Although Hensen's cells responded to every application of TRPA1 agonist, Ca²⁺ responses propagated to the Kolliker's organ upon repeated stimulations (Supplementary Video 2) in most but not all (7 out of 10) regions evaluated. The propagation of Ca²⁺ responses often occurred at specific locations or "hot spots" (Supplementary Fig. 4a, b). Fast-propagating Ca²⁺ waves in the Kolliker's organ (~8 μm/s, Fig. 5a) resembled the previously described Ca²⁺ waves mediated by the release of ATP into the extracellular medium[29,33] and were different from the slow-propagating TRPA1-initiated responses in the Hensen's cells (~0.3 μm/s, Fig. 5a). A non-selective antagonist of ATP receptors, PPADS, inhibited the secondary Ca²⁺ waves in the Kolliker's organ without apparent effects on the Ca²⁺ responses in Hensen's cells and propagation of Ca²⁺ signals from the Hensen's cells to the inner border cells adjacent to the Kolliker's organ (Figs. 5b and 1a). We conclude that TRPA1-evoked Ca²⁺ responses in Hensen's cells are transmitted to the cells of the Kolliker's organ and initiate there the previously described self-propagating ATP-dependent Ca²⁺ waves[34] (Supplementary Fig. 4c). This transmission unlikely involves ATP release from the Hensen's cells and its diffusion to the Kolliker's organ, since the Claudius' cells were not activated (Fig. 5a; Supplementary Fig. 4a), even though they are adjacent to Hensen's cells and very sensitive to extracellular ATP[30].

It is more likely that TRPA1-initiated Ca²⁺ responses propagate beneath the OHCs toward the Kolliker's organ through gap junctions interconnecting supporting cells of the organ of Corti (Supplementary Fig. 4c). This propagation was not easy to observe because the membrane-permeable Ca²⁺ indicator did not usually load the deeply located Deiters' and pillar cells (Supplementary Fig. 5). However, when Deiters' cell loading was successful, we did observe their activation following consecutive Ca²⁺ responses in Hensen's cells (Fig. 5c, d). Ca²⁺ responses in the Deiters' cells were observed even in the presence of PPADS (Fig. 5e, f), which is consistent with an ATP-independent propagation of the signal from Hensen's to Deiters' cells (e.g., through gap junctions). It is not surprising that TRPA1-initiated responses propagate from Hensen's cells across the organ of Corti toward the Kolliker's organ but not toward the Claudius' cells, since several studies demonstrated functionally different gap junctions interconnecting Hensen's cells with Deiters' and Claudius' cells, correspondingly[34,35].

## Deiters' cells express functional TRPA1 channels
Next, we used whole-cell patch-clamp recordings to explore the existence of functional TRPA1 channels in various supporting cells. The electrical coupling between the supporting cells was blocked by flufenamic acid (FFA) since it allowed for more stable recordings than other gap junction blockers tested. Although FFA itself can activate TRPA1 channels[36], we did not observe any FFA-induced changes of

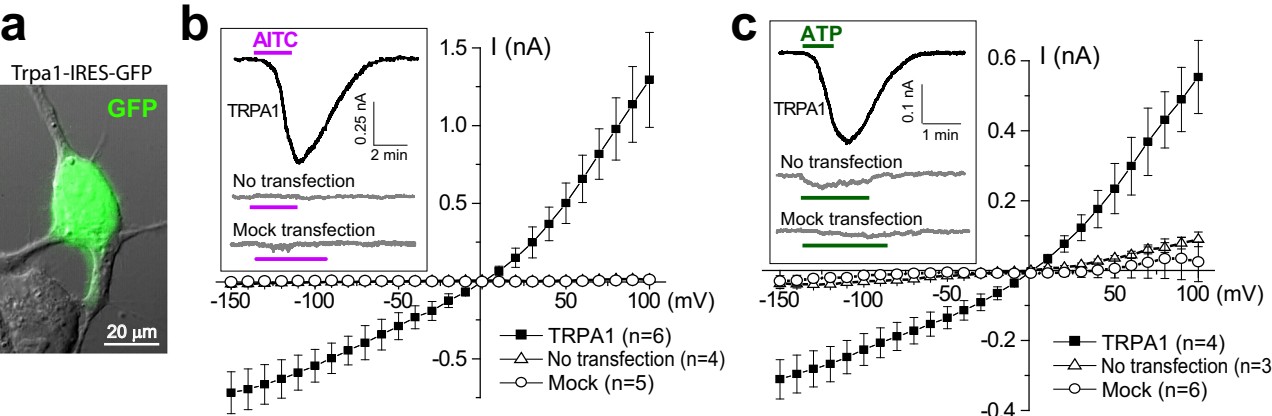

**Fig. 4 | TRPA1 channels can be activated downstream of P2Y receptors.**
**a** Overlay of GFP fluorescence and the bright field image of a HEK293 cell transfected with a *Trpa1*-IRES-*GFP* construct. **b**, **c** Current-voltage relationships of the whole-cell current responses to 20 µM AITC (**b**) and 10 µM ATP (**c**) in HEK293 cells transfected with a *Trpa1*-IRES-*GFP* construct (filled squares), a *GFP*-only construct ("*mock*", open circles) or not transfected (open triangles). The data are shown as Mean ± SE, with n = cells. Insets show representative current traces to the puff stimulation with AITC or ATP. Source data are provided as a Source Data file.

resting $[Ca^{2+}]_i$ in the Hensen's cells upon bath application of 100 µM FFA (124.6 ± 6.3 vs. 116.9 ± 8.8 nM for control and FFA-treated cells, respectively, n = 108/9 and 76/6 cells/regions, $P > 0.4$). In agreement with our $Ca^{2+}$ imaging results, puff application of CA caused a large inward current in Hensen's cells of wild-type but not of *Trpa1*$^{-/-}$ mice (Fig. 5g, j). It is worth mentioning that, in these experiments, we established patch clamp recordings at the apical surface of Hensen's cells without disrupting the epithelium, and therefore we believe that we activated TRPA1 channels at the endolymphatic side of these cells. The TRPA1-dependent current ($I_{TRPA1}$) had a reversal potential close to 0 mV and a prominent inward rectification (Fig. 5g). This rectification is unlikely to result from a potential interaction of FFA with TRPA1, because the same inward rectifying current was evoked by CA in Hensen's cells in the presence of octanol (Fig. 5g), another gap junction blocker that does not interact with mouse TRPA1[37]. Next, a small hole in the epithelium was made to access the basolateral surface of Deiters' and pillar cells. A very similar inward rectifying TRPA1 conductance was also observed in Deiters' cells of wild-type but not of *Trpa1*$^{-/-}$ mice (Fig. 5h, j). Interestingly, CA did not evoke $I_{TRPA1}$ in pillar cells (Fig. 5i, j). We conclude that, in addition to the OHCs that we previously investigated[16], functional TRPA1 channels are present in Hensen's and Deiters' cells. However, these channels seem to be accessible from the endolymphatic side of the epithelium in Hensen's cells but not OHCs, as evidenced by the lack of prominent $Ca^{2+}$ responses in OHCs within the intact undisrupted epithelium (Figs. 2 and 5).

**TRPA1 activation causes tissue movements originating at pillar and Deiters' cells**
Application of TRPA1 agonists to the cochlear epithelium induced not only $Ca^{2+}$ responses (Fig. 2) but also prominent tissue movements (Fig. 6a and Supplementary Video 3). In contrast to the previously described cell shape changes in the Kolliker's organ[23], TRPA1-induced tissue movements were delayed and long-lasting, reminiscent of the $Ca^{2+}$ responses in the Hensen's cells (Fig. 6a, b). At the apical surface of the organ of Corti, these tissue movements typically initiated at the Hensen's cell region and were later seen as pulling toward the region between IHC and OHC (Fig. 6a, b and Supplementary Video 3). Although spontaneous $Ca^{2+}$ waves in the epithelium generated short-lived tissue contractions and complicated the analysis, on average, there was a noticeable increase in tissue movements during and after the TRPA1 agonist application in cochlear explants from wild-type but not from *Trpa1*$^{-/-}$ mice (Fig. 6c, d). Bright field imaging of TRPA1-induced contractions at lower focal planes (near the feet of pillar cells) revealed that, besides Hensen's cells, pillar and Deiters' cells also

exhibit active shape changes (Fig. 6e–g). These movements were observed at P3-P5, before the onset of OHC electromotility at P6[38,39]. Furthermore, we did not observe TRPA1-initiated changes of OHC diameters at this age, even when the TRPA1 agonist had access to the basolateral membrane of OHCs (Fig. 6f and Supplementary Video 4). Thus, we concluded that TRPA1 agonists can evoke substantial tissue displacements and changes to the geometry of the organ of Corti.

**Pillar cells possess $Ca^{2+}$-dependent contractile machinery**
Movements of the phalangeal processes of isolated Deiters' cells were previously observed in response to extracellular ATP[40] or after an increase of $[Ca^{2+}]_i$[41]. However, it was surprising that TRPA1-dependent tissue movements could originate at pillar cells (Fig. 6f, g). Therefore, we investigated whether an increase of $[Ca^{2+}]_i$ could evoke active cell shape changes in pillar cells. We established whole-cell recordings in outer pillar cells with patch pipettes filled with 1 mM of the photo-labile $Ca^{2+}$ chelator, NP-EGTA, pre-mixed with $Ca^{2+}$ ("caged $Ca^{2+}$") and the $Ca^{2+}$ indicator Fluo-4 (Fig. 7a). After ~5 min loading through the patch pipette, a short ultra-violet (UV) laser illumination produced a prominent increase of $[Ca^{2+}]_i$ in cells loaded with caged $Ca^{2+}$ (Fig. 7b–d) but not in control cells where NP-EGTA was replaced with EGTA (Fig. 7c, d). Simultaneous measurements of the diameter of pillar cells at the foot region revealed that UV stimulation caused a slow (~2%/min) decrease of the cell diameter in cells loaded with caged $Ca^{2+}$ but not in control cells nor in any neighboring cells (Fig. 7c, e). Similar to long-lasting non-recoverable changes of $[Ca^{2+}]_i$ after UV stimulation (Fig. 7c, top), the pillar cell contraction was also long-lasting and non-recoverable (Fig. 7c, bottom). The $Ca^{2+}$-evoked changes in pillar cell shape were accompanied with a small drop in the whole-cell current, perhaps, due to activation of $Ca^{2+}$-dependent channels. However, this drop was not different between cells with and without caged $Ca^{2+}$ ($\Delta I_m = -4.29 \pm 2.8$ vs. $-3.23 \pm 0.9$ pA, $P > 0.56$) (Fig. 7c middle). Thus, we conclude that pillar cells possess a $Ca^{2+}$-dependent slow contractile mechanism.

**TRPA1 contributes to the temporary shift of hearing thresholds after noise exposure**
TRPA1-initiated changes of supporting cell shape could alter the geometry and/or stiffness of the organ of Corti, thereby affecting cochlear amplification in vivo. To test the role of TRPA1-dependent responses in cochlear function, we exposed 3–4-week-old mice to moderate broadband (white) noise (100 dB SPL for 30 min). To confirm the generation of endogenous TRPA1 agonists in these experimental conditions, we performed immunolabeling of 4-HNE-modified proteins and found a slowly progressing production of 4-HNE in the

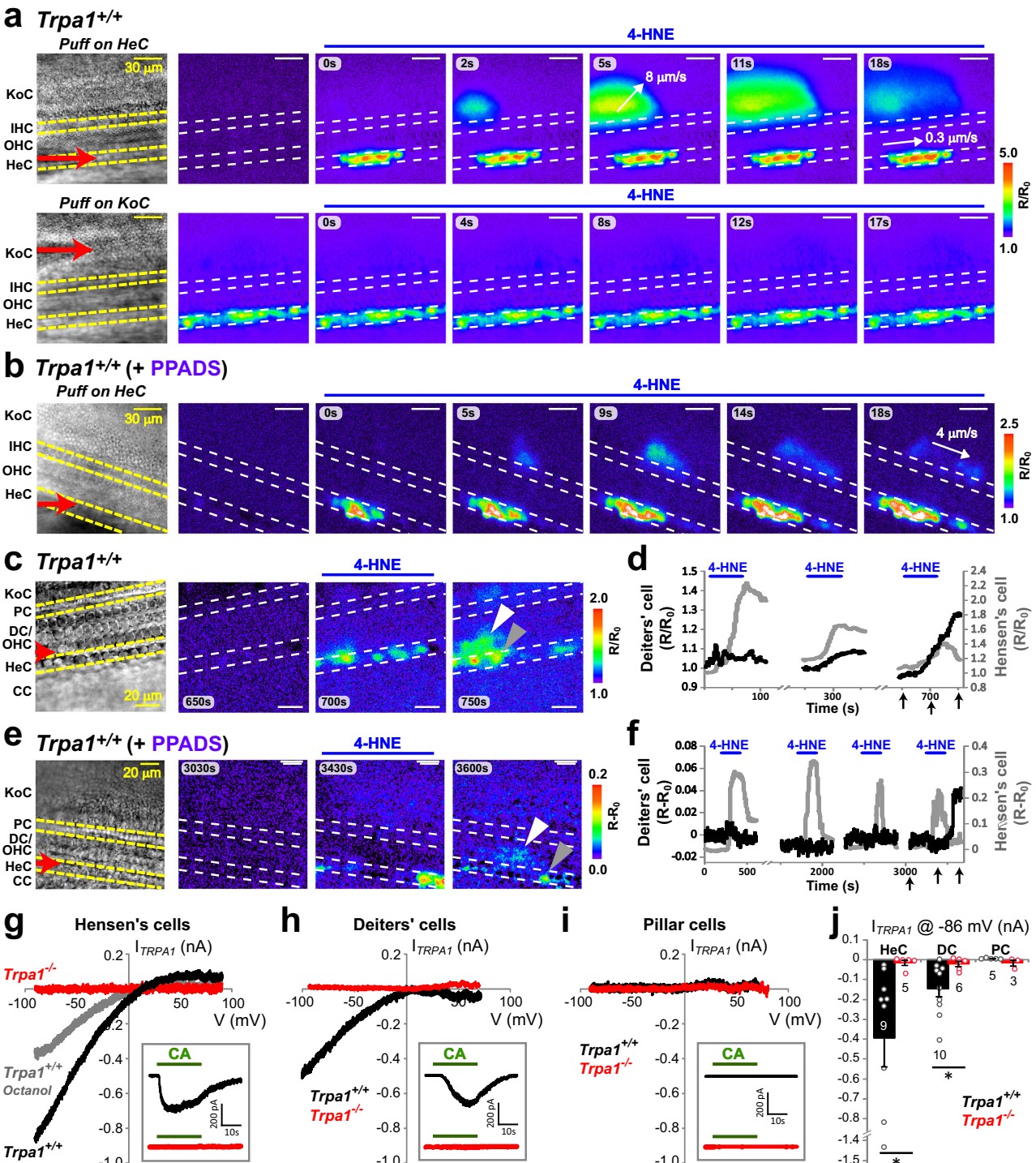

**Fig. 5 | Propagation of TRPA1-initiated Ca²⁺ responses across the organ of Corti.**
**a** Ca²⁺ responses to the local application of 400 μM of 4-HNE in a cochlear explant from a neonatal (P1) wild-type mouse. 4-HNE was locally applied first to the Hensen's cells (top) and then to the cells of the Kolliker's organ (bottom) in the same specimen. The direction of Ca²⁺ response propagation and its speed are indicated by the white arrows and numbers. **b** PPADS (50 μM), an inhibitor of ATP receptors, attenuates Ca²⁺ responses in the Kolliker's organ but not in the Hensen's cells. 4-HNE (400 μM) was applied to a neonatal (P2) wild-type explant. **c**, **e** In an experiment when fura-2 was able to permeate into Deiters' cells, Ca²⁺ responses were observed also in these cells after several puff applications of 200 μM of 4-HNE in the absence (**c**) or presence (**e**) of 50 μM PPADS in the bath solution. Gray and white arrowheads point to the Hensen's and Deiters' cells that were measured in panels (**d**) and (**f**). **d**, **f** [Ca²⁺]ᵢ changes in Deiters' (black) and Hensen's (gray) cells in

the absence (**d**) or presence (**f**) of PPADS. The *x*-axis arrows indicate the timing of frames in (**c**) and (**e**). **g**–**i** Representative current-voltage relationships of TRPA1 conductances activated by CA (100 μM) in wild-type (black and gray) and *Trpa1⁻/⁻* (red) Hensen's (**g**), Deiters' (**h**), and pillar (**i**) cells. Usually, the whole-cell patch-clamp recordings were obtained in the presence of FFA (100 μM) to electrically uncouple the supporting cells, but some Hensen's cells were recorded in the presence of another gap junction blocker, octanol (1 mM) (gray in (**g**)). Insets show whole-cell current responses to CA at holding potential of −80 mV. **j** Average TRPA1-mediated current at −86 mV holding potential in Hensen's (HeC), Deiters' (DC), and pillar (PC) cells in wild-type (black) and *Trpa1⁻/⁻* (red) mice. The data are shown as Mean ± SE. Asterisks indicate statistical significance (*P* = 0.030, HeC; *P* = 0.016, DC; two-sided Student's *t* test). Source data are provided as a Source Data file.

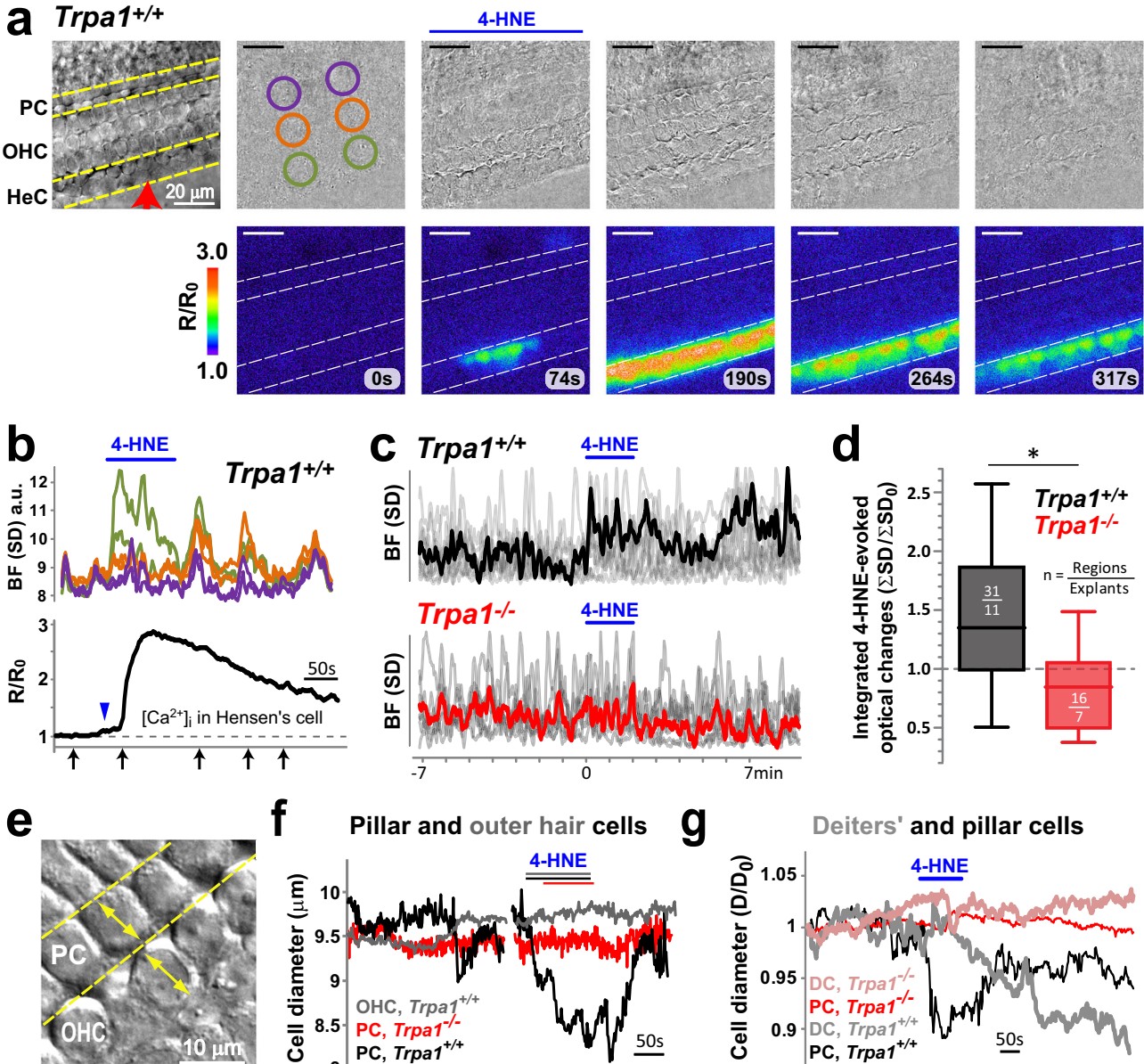

**Fig. 6 | TRPA1 activation evokes prominent displacements in the organ of Corti.**
**a** Tissue movement induced by puff application of 4-HNE. Top, reference bright field image (left) and time-lapse sequence of subtracted bright field (BF) frames taken 10 s apart. The subtraction results in an average gray level if tissue does not move, and in darker or brighter pixels if any movement occurs. Bottom, simultaneous imaging of $Ca^{2+}$ responses. **b** Time course of tissue displacements (top) relative to the $Ca^{2+}$ responses in a Hensen's cell (bottom). Overall tissue displacements were estimated by the average standard deviation (SD) of pixel values within the corresponding color-coded regions of interest circled in (**a**). The x-axis arrows indicate the timing of frames in (**a**). The blue arrowhead points to the initial small rise in $[Ca^{2+}]_i$ upon 4-HNE stimulation that precedes the tissue movement. **c** Superimposed spontaneous and 4-HNE-evoked tissue displacements in 11 wild-type (top) and 7 $Trpa1^{-/-}$ (bottom) cochlear explants, with average traces in bold. **d** The area under the curve for each movement trace in **c** was calculated in identical

time windows (7 min) before and after the beginning of 4-HNE application. The distribution of average fold changes are shown, where a value of one (dashed line) corresponds to no change in overall tissue movement. Boxplots show the median (center line), 25th and 75th percentile (box limits), and minimum and maximum values (whiskers). Asterisk indicates statistical significance ($P = 0.048$, two-sided Student's t test). **e** Bright field image illustrating diameter measurements. **f** Diameter changes of a wild-type outer pillar cell (PC, black) and a neighboring OHC (gray) in response to 4-HNE application. Diameter measurements in a $Trpa1^{-/-}$ outer pillar cell (red) are also shown. **g** 4-HNE-induced changes of cell diameters in Deiters' (DC, light traces) and pillar (PC, dark traces) cells in wild-type (black and gray) and $Trpa1^{-/-}$ (red and pink) cochlear explants. In all panels, the timing of 4-HNE applications is indicated by horizontal bars. All explants were isolated at P3-P4, before the onset of OHC electromotility[38]. Source data are provided as a Source Data file.

cochlea throughout several days after acoustic trauma (Supplementary Fig. 6a–d), exactly as expected from a previous study[28]. Then, we monitored cochlear function with auditory brainstem responses (ABR) and distortion-product otoacoustic emissions (DPOAE), before and immediately after noise exposure, as well as at 5 and 14 days of recovery. Before noise exposure, we observed identical ABR

thresholds (Fig. 8a, inset) and DPOAEs (Supplementary Fig. 7a) in wild-type and $Trpa1^{-/-}$ mice, confirming previous findings[6,7]. In the control wild-type mice, moderate noise (100 dB SPL for 30 min) evoked a prominent increase of ABR thresholds (Fig. 8a) and a complete loss of DPOAEs (Supplementary Fig. 7b) immediately after noise exposure. Two weeks after, ABR thresholds and DPOAEs recovered completely at

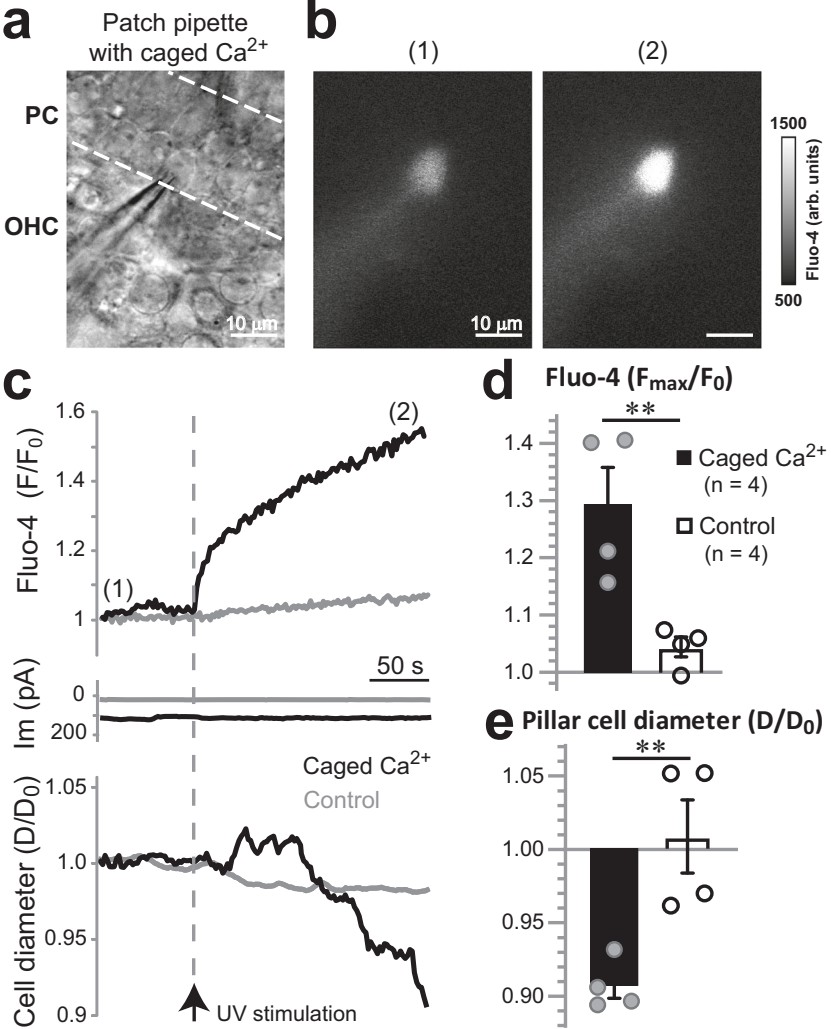

**Fig. 7 | Pillar cells possess Ca²⁺-dependent contractile machinery. a** Bright field image of the experimental setup. An outer pillar cell that was patched with a pipette containing caged Ca²⁺ (1 mM NP-EGTA pre-mixed with 0.7 mM Ca²⁺) and Ca²⁺ indicator, fluo-4 (50 μM). **b** Fluo-4 image of this pillar cell before (1) and after (2) UV flash photolysis (arb. units., arbitrary units). Data are representative of 4 independent series. **c** Representative traces of fluo-4 fluorescence (top), whole-cell current (middle) and the cell diameter (bottom) in the wild-type pillar cell filled with caged Ca²⁺ (black) and a control cell filled with EGTA instead of NP-EGTA (gray). The timing of UV stimulation is indicated with an arrow and a vertical dashed line. **d, e** Average changes of fluo-4 fluorescence (**d**) and cell diameter (**e**) 5 min after UV stimulation in wild-type pillar cells containing either caged Ca²⁺ (filled bars) or control EGTA (open bars). The data are shown as Mean ± SE, with n = pillar cells. Asterisks indicate statistical significance ($P = 0.009$ in (**d**), and $P = 0.008$ in (**e**), two-sided Student's t test). The UV illumination consisted of 15–25 laser pulses (1 ms) for a total duration of 90–150 ms. Source data are provided as a Source Data file.

8 and 16 kHz and only partially at 24 and 32 kHz (Supplementary Fig. 7b). *Trpa1*⁻/⁻ mice showed larger threshold shifts compared to their wild-type littermates immediately after noise exposure (Fig. 8a, $P = 0.008$) but their ABR thresholds recovered faster ($P = 0.002$) (Fig. 8a, b). This faster recovery of hearing thresholds in *Trpa1*⁻/⁻ mice was due to recovery of cochlear amplification as evidenced by the faster recovery of DPOAEs in *Trpa1*⁻/⁻ mice five days after noise exposure ($P = 0.006$) (Fig. 8c). Additionally, in a separate cohort of mice, we evaluated the changes of the latencies of click-evoked ABRs after the same moderate noise exposure. Before noise, the latencies of the ABR wave-I were identical in *Trpa1*⁻/⁻ and control wild-type littermate mice (Fig. 8d, inset). The increase of these latencies was larger in *Trpa1*⁻/⁻ mice immediately after noise exposure but, five days later, the latencies had greater recovery in *Trpa1*⁻/⁻ mice (Fig. 8d). Similarly, ABR wave-I amplitude recovered faster in *Trpa1*⁻/⁻ mice (Supplementary Fig. 7c).

To better explore the threshold shifts occurring during and immediately after noise exposure, we performed an additional set of experiments, in which we measured ABR thresholds only at 16 kHz but

at faster time intervals. As expected, moderate white noise (94 dB SPL, 30 min) produced larger threshold shifts in *Trpa1*⁻/⁻ mice compared to their wild-type littermates (Supplementary Fig. 7d, $P < 0.01$). Thus, TRPA1 channels seem to be functional even immediately after noise exposure, when the byproducts of oxidative stress in the cochlea are yet to be accumulated. This may be explained by TRPA1 activation with extracellular ATP (Fig. 4), which is known to appear in cochlear fluids quite fast after noise exposure[27].

We have also counted IHC ribbon synapses and did not find statistically significant differences between *Trpa1*⁻/⁻ and control mice either before or after noise exposure, even though, two weeks after trauma, we did observe some loss of synapses at high (24–32 kHz) but not at low (8–16 kHz) frequencies (Supplementary Fig. 8). This was consistent with the permanent shift of ABR thresholds after moderate noise only at high frequencies. We concluded that the larger but less prolonged noise-induced temporary shift of hearing thresholds in TRPA1-deficient mice is unlikely to result from IHC synaptic loss but rather represents the loss of TRPA1-mediated effects on cochlear amplification.

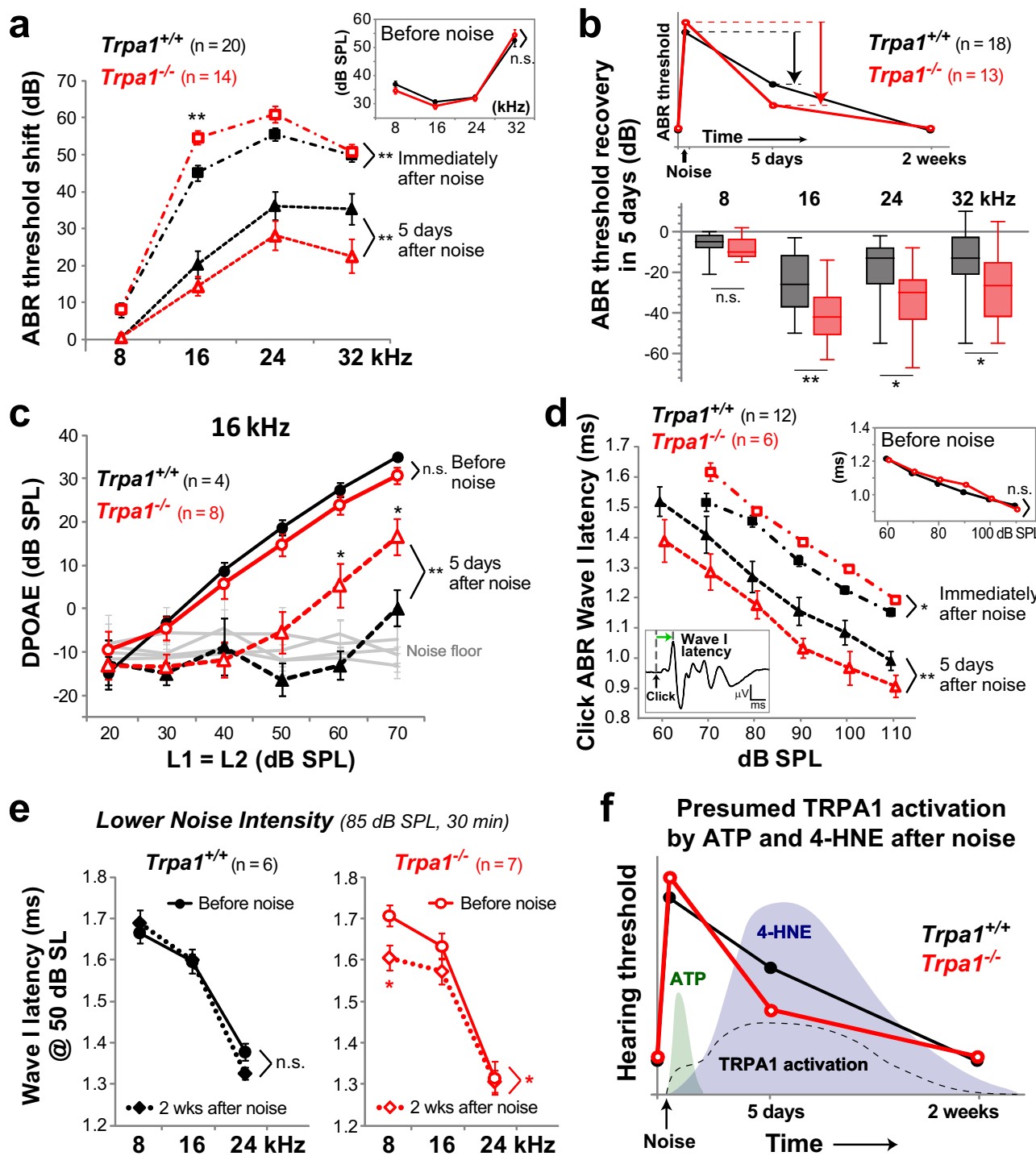

Besides cochlear amplification and DPOAE, TRPA1-initiated changes in the geometry of the organ of Corti may affect stiffness of the cochlear partition and hence the speed of the travelling wave, resulting in non-neural micromechanical changes of the ABR wave-I latency. Therefore, we next explored the effect of a mild noise (broadband, 85 dB SPL for 30 min) that leads to a full recovery of hearing thresholds ($P > 0.12$) (Supplementary Fig. 7e). Two weeks after noise exposure, when the hearing thresholds had fully recovered, $Trpa1^{-/-}$ mice exhibited a permanent decrease in ABR wave-I latency at low frequencies that was not observed in wild-type mice (Fig. 8e), even though there were no differences in wave-I amplitudes at low frequencies (Supplementary Fig. 7f). Thus, TRPA1 channels are likely to protect the cochlea from potentially permanent (or at least two-week-

long) changes in micromechanical properties that otherwise would occur after mild acoustic trauma.

## Discussion

Our study revealed an unexpected function of TRPA1 channels in the cochlea. Similar to TRPA1 in nociceptive neurons, cochlear TRPA1 channels can be activated in stress conditions such as oxidative damage. TRPA1 activation initiates long-lasting $Ca^{2+}$ signaling events that cause active changes of the supporting cell shapes and inhibition of cochlear amplification. These phenomena represent an additional component of the temporary threshold shift after noise exposure that may provide protection after acoustic trauma.

**Fig. 8 | TRPA1 deficiency results in larger but less prolonged inhibition of cochlear amplification after noise exposure. a** Noise-induced shift of hearing thresholds in wild-type (black, filled symbols) and *Trpa1*[−/−] (red, open symbols) mice determined with tone burst-evoked auditory brainstem responses (ABR) immediately (squares, dash-dot lines; difference between the genotypes is significant: ANOVA, *P* = 0.008; post hoc test at 16 kHz, *P* = 0.003) and 5 days (triangles, dashed lines, ANOVA, *P* = 0.002) after exposure to moderate intensity white noise (100 dB SPL, 30 min). The inset shows baseline thresholds in wild-type and *Trpa1*[−/−] mice. **b** Schematic representation of the time course of hearing threshold changes after moderate noise exposure (top) and the actual magnitudes of recovery at 5th day relative to the post-exposure thresholds (bottom). Each boxplot shows the median (center line), 25th and 75th percentile (box limits), and the minimum and maximum values (whiskers). *P* = 0.004 (16 kHz), *P* = 0.02 (24 kHz), *P* = 0.033 (32 kHz). **c** Distortion product otoacoustic emissions (DPOAE) measured before (circles, solid lines) and 5 days after noise exposure (triangles, dashed lines, ANOVA, *P* = 0.006; post hoc tests: *P* = 0.025 and *P* = 0.027 for 60 and 70 dB SPL, correspondingly). **d** Latencies of click-evoked ABR wave-I (as indicated in the left cartoon) before (upper right inset), immediately after (squares, dash-dot lines, ANOVA, *P* = 0.037) and 5 days after moderate noise exposure (triangles, dashed lines, ANOVA, *P* = 0.004). **e** Latencies of tone burst-evoked ABR wave-I at 50 dB sensation level (SL) before (circles, solid lines) and 2 weeks after (diamonds, dotted lines) exposure to the mild (85 dB SPL for 30 min) white noise in wild-type (black) and *Trpa1*[−/−] mice (red, ANOVA, *P* = 0.035; post hoc test at 8 kHz, *P* = 0.025). **f** Schematic diagram of noise-induced changes of hearing thresholds are superimposed to the expected fast ATP release into the cochlear fluids[27] (green shaded area), the delayed generation of 4-HNE (blue shaded area), and the presumed activation of TRPA1 channels in cochlear supporting cells (dashed line). In all panels, data from wild-type mice are black and data from *Trpa1*[−/−] mice are red. Mice were 3–4 weeks old at the time of noise exposure. All data are shown as Mean ± SE. Asterisks indicate statistical significance: *P* < 0.05; **P* < 0.01; ***P* < 0.001; either directly at data points by two-sided Student's *t* test (**a, b, c, e**) or between grouped data by two-way ANOVA (**a, c, d, e**). n.s. not significant. Source data are provided as a Source Data file.

Although TRPA1 channels are expressed in several types of supporting cells in the cochlea, not all of them responded equally to the application of TRPA1 agonists. This is not surprising given that the TRPA1 channels could be activated by many non-covalent ligands and divalent cations and regulated through multiple signaling pathways[4,11,24] including its regulation by intracellular Ca$^{2+}$ (Fig. 3). The most prominent TRPA1-initiated responses were observed in the Hensen's cells. These responses were able to propagate longitudinally and across the organ Corti causing visible tissue displacements in the cochlear epithelium. Most importantly, all of these phenomena were observed upon stimulation with 4-HNE, an endogenous byproduct of lipid peroxidation that is generated in the cochlea during noise-induced oxidative stress[28] (Supplementary Fig. 6). Thus, Hensen's cells are likely to be the major "sensors" of oxidative stress in the cochlea.

Previously, two other mechanisms for "sensing" damage in the cochlea were proposed that involve responses to ATP in the supporting cells[29,31] and in the unmyelinated type II afferent fibers[42]. Although both ATP and TRPA1 agonists evoke Ca$^{2+}$ responses in the cochlear epithelium, there are important distinctions between them. First, TRPA1-initiated Ca$^{2+}$ responses usually persist long after the secession of the stimulus (Fig. 2), which is reminiscent of the long-lasting Ca$^{2+}$ responses evoked by acoustic overstimulation in the pillar and outer hair cells in the guinea pig temporal bone preparation[43,44]. In contrast, ATP-evoked responses are short-lived in each individual cell (Supplementary Fig. 3) even though they can persist in the epithelium for some time as self-propagating Ca$^{2+}$ waves[30]. Second, ATP-evoked Ca$^{2+}$ waves require the activation of P2Y receptors[31,45], while TRPA1-initiated responses propagate across the organ of Corti even when the ATP receptors are blocked (Fig. 4b). Finally, ATP-evoked responses propagate widely throughout the epithelium, while TRPA1-initiated responses are confined within the supporting cells surrounding the outer hair cells and do not typically propagate laterally toward the Claudius' cells (Figs. 2, 4). We do not believe that secondary activation of ATP-dependent Ca$^{2+}$ waves in the Kolliker's organ (Fig. 4a; Supplementary Fig. 4a, b) is relevant to the adult animals, first, because Kolliker's organ disappear/remodel in adults and, second, because it is likely to involve the gap junctional conductance between inner and outer pillar cells that also disappears in adult animals[35].

The specific cellular pattern of TRPA1-initiated responses may have evolved because pillar and Deiters' cells are located underneath the hair cells and have limited access to the apical surface of the cochlear epithelium. Thus, propagation of TRPA1-initiated Ca$^{2+}$ responses from Hensen's cells across the organ of Corti ensures the activation of pillar and Deiters' cells when byproducts of oxidative damage are present in the endolymph (Supplementary Fig. 4c).

Puff applications of 4-HNE to postnatal cochlear explants evoke prominent and long-lasting tissue displacements (Fig. 6). In vivo, these

mechanical events could last even longer since 4-HNE, an endogenous byproduct of lipid peroxidation, is generated in the cochlea throughout several days after noise exposure[28] (Supplementary Fig. 6a–d). Furthermore, TRPA1-initiated movements in the sensory epithelium involve pillar and Deiters' cells, the major cells that determine the cross-sectional geometry of the organ of Corti (Supplementary Fig. 6e). TRPA1-initiated changes in the shape of these cells are expected to modify the geometry of the organ of Corti and alter cochlear amplification. In older animals, these tissue displacements may be supplemented by slow depolarizations and voltage-dependent contractions of OHCs which also express functional TRPA1 channels[16]. Long-lasting contractions of the organ of Corti that involve movements of Hensen's and Deiters' cells have been observed in guinea pig temporal bone preparations after in vitro acoustic overstimulation[46,47]. Furthermore, Liberman's group reported a similar organ of Corti contraction in vivo in the live animals, 24 h after mild noise that produces only TTS[48]. The underlying mechanisms of this phenomenon are still unknown, but our data suggest that TRPA1-mediated signaling may contribute to it. In fact, recent data show that direct optogenetic activation of the Deiters' and outer pillar cells protects the cochlea from desensitization developing during presentation of loud sounds[49]. Hence, it is not surprising that activation of the same cells through propagation of TRPA1 signaling may be also protective.

Elevation of the hearing thresholds immediately after acoustic trauma is likely to result from a combination of cellular damage together with transient protective responses, such as the one associated with the release of ATP into the cochlear fluids[27,50]. TRPA1 channels may enhance the ATP-dependent protection since they can be activated downstream of the P2Y receptors. A decrease in the ATP-TRPA1-dependent protection would allow for greater damage to occur and explain why *Trpa1*[−/−] mice exhibit larger shifts in hearing thresholds immediately after noise exposure. While the release of ATP into cochlear fluids might be an acute phenomenon following noise exposure[27,51], the generation of reactive oxygen species can last several days[28] (Supplementary Fig. 6). During this period, we expect TRPA1 channels to be continuously activated (or re-activated) by lipid peroxidation byproducts such as 4-HNE. Long-lasting TRPA1 activation is likely to maintain a modified geometry of the organ of Corti and contribute to a protective shift in hearing thresholds (Fig. 8f; Supplementary Fig. 6e). In agreement with this hypothesis, *Trpa1*[−/−] mice exhibited a faster recovery of hearing thresholds and cochlear amplification after noise exposure.

Although our data demonstrate the involvement of TRPA1 channels in regulation of cochlear amplification during noise-induced temporary shift of hearing thresholds, we found no differences between wild-type and *Trpa1*[−/−] mice in ABR thresholds or DPOAE amplitudes after recovery from temporary threshold shift,

two weeks after noise exposure. Therefore, it is less clear against which kind of noise-induced changes in the cochlea TRPA1 channels are protecting. Interestingly, despite complete recovery of ABR thresholds, even a very mild noise exposure (85 dB SPL for 30 min) produced in $Trpa1^{-/-}$ mice some permanent (or at least lasting for more than two weeks) changes. While wave-I ABR latencies increased immediately after noise and recovered completely in wild-type mice, they were significantly reduced at low frequencies in $Trpa1^{-/-}$ mice two weeks after noise (Fig. 8e). This decrease in wave-I ABR latency cannot be explained by changes in the number of IHC synapses (Supplementary Fig. 8) and, therefore, may be caused by some permanent changes in the mechanical properties of the cochlear partition that would result in a faster travelling wave and/or a larger pool of activated IHCs. Perhaps, in vivo cochlear imaging with optical coherence tomography techniques is needed to clarify the exact contribution of TRPA1 channels to the noise-induced changes in the micromechanics of the cochlea.

Altogether, our results show that cochlear TRPA1 channels can sense oxidative tissue damage similarly to their counterparts in nociceptive neurons. However, unlike its role in sensory neurons, TRPA1 activation in the supporting cells of the inner ear causes direct morphological changes in the cochlear epithelium and could provide protection after noise exposure by modulating hearing sensitivity.

## Methods

### Mice, cochlear explants, and cochlear whole mounts

TRPA1-deficient mice[7] were kindly provided by Dr. Kwan and Dr. Corey. These mice were backcrossed to C57BL/6 mice (Jackson Laboratories) ten times and sibling breeding pairs were established. Only $Trpa1^{+/+}$ and $Trpa1^{-/-}$ mice were included in this study, because heterozygous ($Trpa1^{+/-}$) mice exhibit an intermediate phenotype on a variety of tests[6,7]. Both male and female mice were included in this study, and experimental groups typically consisted of ~50/50 male/female ratios. The exact numbers of males and females for each experiment are provided in the Source Data file. Mice were housed in Tecniplast PIV cages, in rooms with 14:10 light:dark cycle conditions, at 70 °F and 50% humidity.

Cochlear explants were isolated during early postnatal days (P0–P8). Briefly, young postnatal pups were first cooled down on ice in a glove for about 15 min and then decapitated with sharp scissors. Temporal bones were dissected from the skull and placed in cold Leibovitz's L-15 medium (Gibco), the cochlear wall was carefully removed with fine forceps, and the underlying soft tissue was carefully pulled away from the modiolus (starting at the base of the cochlea). The explanted tissue included the spiral limbus, inner sulcus, organ of Corti and, in some experiments, outer sulcus, spiral prominence and stria vascularis. In explants were the stria vascularis was not removed, the Reissner's membrane was carefully torn using fine forceps. The cochlear explants were held in place by two glass fibers glued to the bottom of a glass-bottom Petri dish. Explants were used either immediately after dissection or kept in DMEM (Gibco) supplemented with 7% fetal bovine serum (FBS, Atlanta Biologicals) and 10 μg/mL ampicillin (Calbiochem) at 37 °C and 5% $CO_2$ for 1–3 days. All experiments with cochlear explants were performed in the middle region of the cochlea and at room temperature.

Adult (>P17) mice were euthanized in $CO_2$ chamber and then decapitated. The dissected cochleae were glued to the bottom of a Petri dish using cyanoacrylate plastic bonder. The cochleae were submerged in Leibovitz's L-15 medium (Gibco) and the apical bone, stria vascularis and Reissner's membrane were carefully removed to allow the delivery of drugs and imaging.

All animal procedures were approved by the University of Kentucky Animal Care and Use Committee (protocols 00903M2005, 2019-3414 and 2020-3535).

## Immunolabeling

**PLAP immunolabeling.** Postnatal organ of Corti explants were fixed in 4% paraformaldehyde (PFA, Electron Microscopy Sciences) in 1× PBS for 1–4 h (h) at 4 °C, rinsed/permeabilized in a wash solution containing PBS and Triton-X (0.01 or 0.1%), and incubated in blocking solution consisting of 10% normal goat serum (NGS) in the wash solution. PLAP (clone 8B6) antibody (Sigma-Aldrich, catalog # A2951) was added (1:50 to 1:100) to the blocking solution and incubated overnight at 4 °C. Samples were rinsed with the wash solution and incubated with the fluorescently labeled goat anti-mouse Alexa Fluor 488 or 568 secondary antibodies (1:1000, ThermoFisher Scientific, catalog # A-21131 and A-11004, respectively) for 2 h at room temperature. The samples were counterstained with fluorescent phalloidin (1–2 units), Hoechst 33235 (500 ng/mL) or DAPI (300 nM) at room temperature for 0.3–2 h. Cochleae were rinsed in PBS and mounted in Prolong Gold Anti-Fade. Reagents for immunostaining were obtained from Invitrogen unless otherwise stated. Images were acquired with an Olympus FluoView 1000 confocal microscope using a 63× PlanSApo objective with numerical aperture (NA) of 1.42 or an Olympus BX51WI microscope with 100× lens (LUMPlanFL, 1.0 NA) and DSU spinning disc confocal attachment.

**HNE and CtBP2 immunolabeling.** Temporal bones from control and noise-exposed 4–5-week-old mice were dissected, and the cochleae were gently perfused with 4% PFA in 1× PBS through small openings in the apical and basal turns. Temporal bones were kept in fixative at 4 °C overnight, then rinsed with PBS and placed in a 0.1 M EDTA solution in water (Sigma-Aldrich) for 48 h. Temporal bones were rinsed in PBS and each cochlea duct was dissected into 4 regions. For HNE labeling, the tissue was permeabilized/blocked in a solution containing 1% Triton-X and 10% NGS in PBS for 30 min and then incubated with an anti-HNE-Michael adducts rabbit polyclonal antibody (1:150 dilution, Abcam, catalog # ab46545) in a 10% NGS, 0.1% Triton-X solution at 4 °C for 4 days. For CtBP2 labeling, the tissue was permeabilized with 0.5% Triton-X for 1 h, blocked with 10% NGS in PBS for 1 h, and then incubated with an anti-CtBP2 mouse IgG1 monoclonal antibody (1:200 dilution, BD Biosciences, clone 16, catalog # 612044) overnight in blocking solution at 4 °C. Samples were washed and incubated with a fluorescently labeled secondary antibodies (1:200 goat anti-rabbit DyLight 488, Jackson Immunoresearch, catalog #111-485-144; or 1:1000 goat anti-mouse IgG1 Alexa Fluor 555, ThermoFisher Scientific, catalog # A-21127) overnight at 4 °C. Mounting was performed as described above. Images were acquired with an Olympus BX51WI spinning disc confocal microscope (60× LUMPlanFL, 1.0 NA) or a Nikon Eclipse TE2000-U inverted microscope equipped with a spinning-disk confocal unit (100× TIRF, 1.49 NA).

## Drug delivery

Cells were stimulated with local application of 200-400 μM of 4-HNE (Cayman Chemical, catalog # 32100), 100-200 μM of CA (Sigma-Aldrich, catalog # W228605), 2–250 μM of ATP (Sigma-Aldrich), 20–100 μM of AITC (Sigma-Aldrich, catalog # 377430) or 5-200 μM pBQN (Sigma-Aldrich, catalog # B10358) through puff pipettes. The size of the stimulated area was regulated by controlling the diameter of the pipette tip (3–6 μm) and the pressure applied to the puff pipette (2–4 kPa). The pipette tip was positioned 8–10 μm above the surface of the cells to be stimulated. Pressure pulses were generated by a pneumatic pico-pump (World Precision Instruments) that was triggered by MetaMorph or pClamp software (Molecular Devices). The output of the pico-pump was also coupled to a Traceable™ manometer gauge (Fisher Scientific), which allowed the precise measurement and adjustment of the applied pressure. The system had a small delay (~40 ms) between the software-generated trigger and the beginning of the actual drug delivery to the cells. This delay was determined by placing a puff pipette at a distance of 8–10 μm from the patch-clamp

pipette and measuring the shift of the patch-clamp pipette offset upon puff application of a K⁺-rich solution. We quantified Ca²⁺ responses only in the cells that were within a 35 μm radius around the tip of the puff pipette.

### Ratiometric calcium imaging

Cells were loaded with the cell-permeable (AM) version of the Ca²⁺ indicator fura-2 (Molecular Probes, catalog # F1221) at room temperature in L-15 medium. Cochlear explants were loaded at a concentration of 10 μM for 50 min and cochlear whole mounts at 20 μM for 20 min. The fura-2-AM was premixed with a 20% solution of Pluronic F-127 in DMSO (Molecular Probes) to improve cell permeabilization. The loaded tissues were rinsed with L-15 medium and imaged 20 min later (to allow for the complete cleavage of intracellular AM esters). In some experiments, 50 μM pyridoxalphosphate-6-azo-phenyl-2',4'-disulfonic acid (PPADS) was applied to the bath solution during imaging to block ATP receptors. A Ca²⁺ calibration kit (Molecular probes) was used to calculate intracellular Ca²⁺ concentrations $[Ca^{2+}]_i$ based on the ratio of fura-2 signals evoked by 340 ($F_{340}$) and 380 ($F_{380}$) nm illuminations.

Imaging was performed using an upright Olympus BX51WI microscope equipped with 40× (0.8 NA), 60× (1.0) or 100× (1.0 NA) LUMPlanFL water-immersion objectives or an inverted Eclipse TE2000-U Nikon microscope equipped with a 100× (1.3 NA) oil-immersion objective. Fast switching between illuminations at 340 and 380 nm was performed with Lambda DG-4 (Sutter Instrument). Alternating $F_{340}$ and $F_{380}$ image pairs were obtained at acquisition rates of 0.3–2 image pairs per second. The photobleaching of fura-2 dye was estimated to be <0.04% per second. To avoid photo-toxicity from ultra-violet (UV) illumination during prolonged recordings, imaging was often paused at different time points (e.g., during washout periods) which resulted in the discontinuity of $[Ca^{2+}]_i$ traces in some experiments.

### Laser uncaging

The output of a 355 nm diode-pumped solid-state laser (DPSL-355/30, Rapp OptoElectronic) was coupled to an upright Olympus BX51W1 microscope to provide a UV spot illumination of ~10 μm in diameter. The photo-liable Ca²⁺ chelator, o-nitrophenyl ethylene glycol tetra-acetic acid (NP-EGTA, Molecular Probes, catalog # N6802), pre-mixed with Ca²⁺ was loaded into pillar cells through the patch pipette. Cells were patched using pipettes with a resistance of 6–10 MΩ when measured in the bath. The intracellular solution contained (in mM): KCl (12.6), KGlu (131.4), CaCl₂ (0.7), K₂HPO₄ (8), KH₂PO₄ (2), Na₄-ATP (2), Na₄-GTP (0.2), NP-EGTA (1), and fluo-4 (0.05, Molecular Probes). The flash photolysis of NP-EGTA-Ca²⁺ complex was achieved with a train of 15–25 laser pulses of 1 ms duration delivered at 166 Hz. The total duration of the pulse train was 90–150 ms. The optimal number of pulses was determined empirically to generate maximal $[Ca^{2+}]_i$ increase but minimize photobleaching of the Ca²⁺ indicator. Simultaneous time lapse bright-field and epifluorescent imaging were performed. The L-15 medium in the bath was supplemented with 100 μM of FFA to block the gap junctions between the supporting cells. Pillar cells were held at their resting potentials: −30 to −40 mV. In the control cells, the NP-EGTA in the intracellular solution was replaced with the equivalent amount of EGTA.

### Quantification of changes in cell diameters

Widths or diameters of the pillar and Deiters' cells were measured near the base or foot of each cell using algorithms originally designed for the measurements of the outer hair cell electromotility with the accuracy of ~20 nm[52]. These algorithms are archived in Zenodo.org with the identifier (https://doi.org/10.5281/zenodo.7896132)[53]. Using the MetaMorph software, the intensity profiles were obtained in each frame of the time-lapse sequence along a line that was placed across the cell of interest. Next, the movements of the cell edges were quantified with custom scripts in MATLAB (MathWorks). The cell diameter of the patched cell was normalized to the diameter of a neighboring "untouched" cell to account for the artifacts of the small drifts in the focal plane.

### Quantification of TRPA1-dependent tissue displacements

Time-lapse bright field imaging was performed with or without simultaneous fura-2 imaging using an upright Olympus BX51W1 microscope equipped with a 100× LUMPlanFL 1.0 NA objective and differential interference contrast (DIC) optics. To highlight the areas of tissue movements, we subtracted pairs of frames that were 10 s apart. This time shift was determined empirically to emphasize TRPA1-initiated tissue movements. Subtracted frames constitute a new time-lapse sequence that lacked the actual image of the cells but showed only the movements of the cell edges. The standard deviation of the grayscale pixel values within an area of interest in these frames was used as a semi-quantitative measure of tissue displacements. Unfortunately, a more precise "optical flow" analysis of the tissue movement was not effective because pixels at the edge of the cell were hard to track. In DIC imaging, the changes of cell shape often led to prominent alterations in light scattering at the edges.

### TRPA1-dependent ion currents in supporting cells

Ion currents evoked by the puff application of TRPA1 agonists were studied in conventional whole-cell patch-clamp recordings using pCLAMP and Clampfit software. Patch pipettes with a resistance of 5–12 MΩ as measured in the bath were used. The intracellular solution contained (in mM): KCl (12.6), KGlu (131.4), MgCl₂ (2), EGTA (0.5), K₂HPO₄ (8), KH₂PO₄ (2), Mg₂-ATP (2) and Na₄-GTP (0.2). The L-15 medium in the bath was supplemented with 100 μM of FFA (Sigma-Aldrich, catalog # F9005) or 1 mM Octanol (Sigma-Aldrich). Voltage ramps from −90 mV to +90 mV over 300 ms were applied to study the current-voltage relationships before, during, and 5 min after (washout) puff stimulation with CA. Each type of supporting cell was held close to its resting potential. In the presence of FFA, these resting potentials were found to be (in mV): Hensen's cells (0 to +10), Deiters' cells (−10 to −20), outer pillar cells (−20 to −30), and inner pillar cells (−40 to −50).

### TRPA1 activation in heterologous cells

HEK293 cells (ATCC) were cultured on glass-bottom dishes in DMEM supplemented with 7% FBS and 10 μg/mL ampicillin at 37 °C and 5% CO₂. When the cells reached 70% confluence, they were transfected using Lipofectamine 2000 (Invitrogen). Cells were transfected with a bicistronic construct expressing AcGFP1 and FLAG-mouse *Trpa1*[16] or with a control construct expressing only AcGFP1. The transfected cells were identified by monitoring the epifluorescent GFP signal and patched using pipettes with a resistance of 2–5 MΩ as measured in the bath. The intracellular solution contained (in mM): CsCl (147), MgCl₂ (2.5), EGTA (1), Na₂-ATP (2.5) and HEPES (5) and the extracellular solution consisted of HBSS supplemented with CaCl₂ and MgCl₂ to achieve final concentrations of 6.26 mM Ca²⁺ and 2 mM Mg, respectively. HEK293 cells were held at −60 mV and the current-voltage relationships before, during, and after puff stimuli were studied using voltage ramps from −150 to +100 mV.

### Auditory brainstem responses (ABR) and distortion product otoacoustic emissions (DPOAE)

Young adult (3-4 week old) mice (roughly 50:50 ratio of male:female mice in each group) were anesthetized with intraperitoneal injections of 2,2,2-Tribromoethanol (Avertin, 0.4 mg/g of body weight) (Sigma-Aldrich, catalog # T48402). ABR and DPOAE were measured using the System 3 Auditory Workstation equipped with the BioSigRP software (Tucker-Davis Technologies). Frequency-specific tone bursts (3 ms) or click stimuli (33 μs) with alternating polarity were presented at 20 Hz

through an MF-1 speaker (Tucker-Davis Technologies) while signals from subdermal needle electrodes were averaged 512–1024 times to reveal ABRs. To measure DPOAEs, a low-noise microphone (ER-10B+ model, Etymotic Research, Inc.) was placed in the ear canal to measure the intensity of the $2f_1$-$f_2$ response while two electrostatic speakers (EC1 models, Tucker-Davis Technologies) delivered $f_1$ and $f_2$ frequencies at a 1.2 ratio with equal sound pressure levels (SPL). Mice were exposed to white noise with 100 dB SPL (moderate) or 85 dB SPL (mild) intensity for 30 min. ABR and DPOAE measurements were performed before and at several time points after the noise exposure. ABR waveform measurements were performed in Origin and Microsoft Excel.

### Statistical analysis
Results between wild-type and $Trpa1^{-/-}$ mice were compared, and statistical significance was assessed using two-way ANOVAs and two-sided Student's $t$ tests in Microsoft Excel and GraphPad Prism. Results are shown as mean ± standard error (SEM), unless otherwise noted. Asterisks denote statistically significant differences with $P$ values < 0.05 (*), 0.01 (**), 0.001 (***), and 0.0001 (****).

### Reporting summary
Further information on research design is available in the Nature Portfolio Reporting Summary linked to this article.

## Data availability
Hearing testing records and all time-lapse and 3D imaging stacks generated during the current study are available from the corresponding authors on request. Source data are provided with this paper.

## Code availability
The MATLAB scripts to measure changes in cell diameter are archived in Zenodo with the identifier (https://doi.org/10.5281/zenodo.7896132)[53].

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

## Acknowledgements

We thank Dr. David P. Corey and Dr. Kelvin Y. Kwan for providing us the *Trpa1*⁻/⁻ mice and the initial PLAP immunostaining images; Erich T. Boger and Dr. Thomas B. Friedman for providing us with the bicistronic GFP/TRPA1 construct; Dr. Timothy S. McClintock, Dr. Bradley K. Taylor, Dr. Kenneth S. Campbell, Dr. Olivier Thibault, and Dr. Douglas A. Andres for helpful comments and suggestions; Kathryn M. Toner for help with ABR recordings; and Dr. Brandon K. Fornwalt and Dr. Jonathan Suever for guidance on optical flow analysis. The study was supported by NIDCD/NIH (DC009434 and DC014658 to G.I.F.), NIH Shared Instrumentation Grant (S10RR027400 to B.K. Taylor), by the Action on Hearing Loss foundation (G40 to G.I.F.), and by the Hearing Health Foundation (2018 ERG to A.C.V.).

## Author contributions

A.C.V. and G.I.F. designed the study, analyzed and interpreted the data, and wrote the manuscript. A.C.V. carried out Ca²⁺ and bright field imaging experiments, Ca²⁺ uncaging, HNE immunolabeling, PLAP immunostaining, and performed all quantitative data analysis. S.T. and A.C.V. performed the immunolabeling of synaptic ribbons. R.S. carried out patch-clamp recordings in transfected HEK 293 cells and initial ABR recordings. D.A.M. and J.S.N. performed some of the DPOAE recordings. A.C.V. and S.E.E. performed all other in vivo experiments. A.C.V. and C.P. acquired the patch-clamp recordings in cochlear supporting cells. G.P.S. optimized the intracellular solutions used for the laser uncaging experiments. All authors discussed the results and contributed to the final version of the manuscript.

## Competing interests

The authors declare no competing interests.
