## [Peer Review File · Nature Communications]

TRPA1 activation in non-sensory supporting cells contributes to regulation of cochlear sensitivity after acoustic traumaEditorial Note: This manuscript has been previously reviewed at another journal that is not operating a transparent peer review scheme. This document only contains reviewer comments and rebuttal letters for versions considered at *Nature Communications*.

Reviewers' Comments:

Reviewer #1 (Remarks to the Author):

In "TRPA1 channels regulate cochlear amplification through active shape changes of supporting cells in the inner ear" the authors present a detailed investigation of cochlear expression of TRPA1 and demonstrate a novel physiological role for the channel. The authors suggest that TRPA1 activation by lipid peroxidation products formed as a consequence of noise exposure, stimulate contraction and shape changes in Deiter's and pillar cells. This shape change in turn regulates cochlear amplification. This is the first demonstration of TRPA1 regulating cellular shape or producing contraction and it is the first demonstration of TRPA1 influencing hearing. The manuscript demonstrates TRPA1 expression histochemically in several cell types in the cochlea, but it was not possible to demonstrate functional TRPA1 expression in all types of TRPA1 positive cells. Although the discrepancy between TRPA1 expression and TRPA1 function doesn't invalidate the main conclusions of the manuscript, it may distract the reader from the positive findings. The discussion of a number of different cell types in the manuscript may present a barrier to scientists outside the immediate auditory/cochlear field.

Specific comments.

Flufenamic acid potentiates Hensen's cells responses to 4-HNE, but the discussion around TRPA1 requiring synergistic stimulation for activation in Claudius cells is not coherent. A requirement for concerted actions by different agonists has not been suggested or demonstrated for TRPA1 before.

Flufenamic acid can stimulate TRPA1, but can also act on several other ion channels. Data demonstrating that the response evoked by 4-HNE in the presence of FFA can be inhibited by a TRPA1 antagonist would strengthen the conclusion from this experiment. Whereas there may be many possible explanations for why PLAP expression from the TRPA1 promoter in the *Trpa1*^{-/-} does not correlate precisely with functional TRPA1 in the *Trpa1*^{+/+}, the FFA results do not show that TRPA1 activation in the relevant cell types require "proper integration of several signals".

A vast number of reactive molecules covalently modify and activate the channel. Many non-covalent ligands and divalent cations (in particular Ca²⁺ and Zn²⁺) also activate TRPA1 but what are the regulatory mechanisms the authors refer to in the discussion, 2nd paragraph, line 3?

Annexin A2 has been shown to limit the cellular pool of TRPA1 present in the membrane (the only negative modulator identified so far?). Are the authors suggesting a similar mechanism here?

4-HNE is formed endogenously along with many other lipid peroxidation products and radicals under oxidative conditions. However, 4-HNE is not the most potent of TRPA1 agonists and in attempting to demonstrate the presence of functional TRPA1 channels, the authors may be better off using a more potent and more rapidly acting tool compound such as e.g. p-benzoquinone.

TRPA1 antibodies generated non-specific staining: Is this staining non-specific or could it be an indication that some non-functional protein is made in the *Trpa1*^{-/-} mice?

Discussion, first paragraph; the auditory nerve doesn't transmit nociceptive information, since auditory information is not normally transmitted to higher centres relevant for pain. The identity of the transduction molecule responsible for action potential firing doesn't define what sensory modality is transmitted.

The authors observe marked uptake of FM1-43 in OHC, but no Ca²⁺-responses evoked by TRPA1

agonists in the same cells. In contrast, Hensen's cells, which do display TRPA1 mediated Ca²⁺-influx responses, show very little evidence of FM1-43 uptake. These results do not support FM1-43 entry through TRPA1. These results rather seem to suggest that FM1-43 has labelled vesicles in OHC (the conventional use for FM1-43 and related styryl dyes), rather than entered the cells via TRPA1.

Results Page 6, 2nd paragraph: The temporal pattern of the Ca²⁺-responses evoked by 4-HNE in the Kolliker's organ is very different from that observed in Hensen's cells. This is not sufficient to conclude that they are not mediated by TRPA1 activation. Near the bottom of page 8, the explanation for this response in Kolliker's organ is provided.

Reviewer #2 (Remarks to the Author):

Previous studies from the authors have shown that TrpA1 is expressed in the inner ear and can regulate the uptake of aminoglycosides and FM1-43 into cochlear outer hair cells. However, the cells that express TrpA1 in the inner have not been determined without ambiguity and the physiological function of TrpA1 in the inner still needs to be determined. In the current manuscript, the authors therefore further characterize TrpA1 expression and function in the inner ear. Using knock-in mice expressing PLAP from the TrpA1 genetic locus as well as TrpA1 knock-out mice and pharmacological approaches, they provide evidence that TrpA1 is expressed in several inner ear cell types such as hair cells, Hensen's cells, Deiter's cells and Claudius cells as well as cells in Kolliker's organ. Activation of TrpA1 in inner ear organ cultures using pharmacological approaches leads to changes in endogenous Ca²⁺ levels in Hensens cells that appear to be propagated to Kolliker's organ. Ca²⁺ levels also appear to change in several other cell types. TrpA1 agonists also induce shape changes in the organ of Corti. TrpA1 mice appear to be more sensitive to noise trauma when analyzed shortly after noise exposure but seem to recover somewhat better from noise trauma indicative of a protective function for TrpA1 in stressful conditions. The authors propose a link between TrpA1, the shape changes and protection from damage.

The manuscript is interesting but also has significant drawbacks. The authors conclude that they have identified a new mechanism that regulates cochlear sensitivity that depends on TrpA1. They conclude that endogenous noise-induced compounds activate TrpA1, which leads to shape changes in the cochlea and inhibits cochlear amplification. They argue that TrpA1 in Hensen's cells have a central function in this process. These are far-reaching conclusion that are not supported by direct experimental evidence. From the data in the manuscript it is not entirely clear which cells express the functionally relevant TrpA1 receptor. Gene expression studies (PLAP knock-in) do not seem to agree with the results obtained by identifying cells expressing TrpA1 using pharmacological activators. Several cell types seem to respond to TrpA1 agonists and while the signal in Hensen's cells is particularly strong, other cell types also show a significant response. While pharmacological activation of TrpA1 leads to shape changes in inner ear organ cultures, the extent to which this is relevant to the observed protective effect to noise in vivo remains speculative. It is not clear whether noise induces TrpA1 function in vivo and causes shape changes within the organ of Corti. There is also no direct link between the shape changes and protection. Thus the results are correlative and do not show causality. The link to cochlear amplification is far fetched and not supported by experimental evidence. As such, the manuscript neither clearly defines the function of TrpA1 in the inner ear nor identifies a new mechanism that regulates cochlear sensitivity. Thus, while the findings are certainly worthwhile reporting, they seem to be suitable for a more specialized journal.

Reviewer #3 (Remarks to the Author):

Velez-Ortega et al., "TRPA1 channels regulate cochlear amplification through active shape changes..."

In this paper the authors provide evidence that products of oxidative stress act on TRPA1 channels to alter cochlear mechanics and sensitivity over the course of several days. The majority of experiments are convincing and the significance of the findings high. Temporary loss of sensitivity (TTS - temporary threshold shift) is a common consequence of acoustic trauma and this work provides novel insights into underlying molecular mechanisms. The authors are to be commended for the extensive documentation they provide - the main body of the paper is elegantly illustrated and clearly described. However, a general problem is that the eight supplemental figures are provided with far less depth of description and rigor than the eight figures that make up the main document. For example, no statistical treatment is given for results reported in Supplemental Figures. At the same time, these provide anecdotal support for a number of interesting findings. If properly described and quantified these could constitute another interesting and strong publication in its own right. Perhaps the temptation was to add as much support as possible to the main conclusions. However, there is a cost in the struggle imposed on the Reader to weave together a fragmented presentation with varying levels of proof.

Some trimming and/or editing also would help the supplementary videos. Video #1 for example is unnecessary since video #2 incorporates the same point (and is by far the most impressive and useful of the supplementary materials). Video #3 would be much improved with labels to help the viewer identify cell types. Video #4 was quite confusing. In addition to also needing labels, it appears that movement precedes, as well as follows, the appearance of the white arrow. Which is the significant bit?

TrpA1 null mice have greater initial loss of sensitivity, but more rapid recovery after acoustic trauma. From this latter effect the authors draw the somewhat surprising conclusion that longer-term TrpA1-mediated changes are protective to the cochlea. Isn't temporary threshold shift a sign of cochlear damage? If longer-lasting in TRPA1 wild-types then one ordinarily would conclude there was more damage??? A final general point is that all their ex vivo studies examine cellular changes over the course of seconds to minutes. However from this they extrapolate to a days-long effect after acoustic trauma. They suggest that production of TrpA1 agonists continues for days, and show immunolabeling experiments in support of that proposal. This last supplemental figure is a good example of the general problem mentioned above. There is no critical analysis of these results, no mention of number of biological repetitions, and scant description of the methods or controls employed (e.g., "The specificity of the anti-HNE antibodies was tested in control experiments where we exposed postnatal cochlear explants to increasing concentrations (0, 0.02, 0.2 and 2 mM) of H₂O₂ (Invitrogen) for 4 to 6 h."). What else might the antibody bind to in treated tissue?

It should be emphasized that the above remarks don't negate the main significance and substance of this paper which will make an important contribution to the study of cochlear pathogenesis. Several additional suggestions follow.

Abstract:

"...novel mechanism for regulating hearing sensitivity after acoustic trauma..." why not a novel mechanism for temporary threshold shift?

Introduction:

"Although the inner ear is believed to have no nociceptors..." but see recent publications such as

Flores EN, Duggan A, Madathany T, Hogan AK, Márquez FG, Kumar G, Seal RP, Edwards RH, Liberman MC, García-Añoveros J. A non-canonical pathway from cochlea to brain signals tissue-damaging noise. *Curr Biol.* 2015 Mar 2;25(5):606-12. PMID: PMC4348215.

"TRPA1 null mice exhibit normal hearing" - so is proposed protective effect of TRPA1 not evident?

Results:

PLAP under TRPA1 promoter - what was positive control? What was negative control? Would be helpful to have some independent evidence that the promoter construct actually drove specific expression - if possible.

Rise of intracellular calcium very slow. Can this be due to influx across the plasma membrane? If so, how can it take such a long time (38 seconds)?

"Despite the high frequency of spontaneous Ca waves in the Kolliker's organ..." How about different wording here? "Although spontaneous calcium waves occurred frequently in Kolliker's organ, nonetheless it was possible to correlate 4-HNE exposure with the triggering of calcium waves.

"...release of ATP into the extracellular medium..." check references for this statement - are these the correct ones?

Heterologous expression in HEK cells. While this is an interesting experiment in its own right, it is not conclusive (no proof that ATP acts through the TRPA1, simply correlated with expression, and no test of signaling pathway), and not proof of anything in cochlear cells. This figure could be left out.

"...no obvious alterations in whole-cell current...which argues against cell volume changes due to ion fluxes..." What about loss of KCl (charge neutral)? One or the other ion specifically-gated, the other follows passively. The time course of change is certainly slow enough.

Discussion:

"TRPA1 channels do not generate specific nociceptive responses in auditory nerve fibres." How do you know this? Were recordings made showing no increased activity in type I or type II afferents?

"...causing physiologically significant tissue displacements..." Justified statement? How about just 'visible' displacements?

"...in older mice because it is likely to involve gap junctional conductance between inner and outer pillar cells that disappears in adult animals." Not to mention that Kolliker's organ disappears!

Figure 1. specify age illustrated in A. Specify z-axis illustrated in B -orthogonal to plane of previous image?

Figure 6. Put labeled axis on panel 'c' to show where 7 minute integral was collected for panel 'd'. Otherwise the eye is drawn just to the time of drug application and the effect is not convincing.

Figure 6e. which diameter measured - at what point along cell?

Figure 7b. specify time after UV flash that image was taken.

We thank the Reviewers for their thoughtful comments that prompted us to perform a number of additional experiments. Although these experiments were not able to clarify all the details of TRPA1-mediated signaling in the cochlea, they confirmed our major conclusion that TRPA1 channels are involved in the temporary shift of hearing thresholds after acoustic trauma (TTS) and provided a potential mechanism for this involvement. As of today, the cellular and molecular mechanisms of TTS are largely speculative. Therefore, we believe that our study is an important contribution to the fields of the cochlear biology and the sensory neuroscience, in general. The point-by-point answers to the Reviewer's comments are given below.

Reviewer #1:

In "TRPA1 channels regulate cochlear amplification through active shape changes of supporting cells in the inner ear" the authors present a detailed investigation of cochlear expression of TRPA1 and demonstrate a novel physiological role for the channel. The authors suggest that TRPA1 activation by lipid peroxidation products formed as a consequence of noise exposure, stimulate contraction and shape changes in Deiter's and pillar cells. This shape change in turn regulates cochlear amplification. This is the first demonstration of TRPA1 regulating cellular shape or producing contraction and it is the first demonstration of TRPA1 influencing hearing.

The manuscript demonstrates TRPA1 expression histochemically in several cell types in the cochlea, but it was not possible to demonstrate functional TRPA1 expression in all types of TRPA1 positive cells. Although the discrepancy between TRPA1 expression and TRPA1 function doesn't invalidate the main conclusions of the manuscript, it may distract the reader from the positive findings. The discussion of a number of different cell types in the manuscript may present a barrier to scientists outside the immediate auditory/cochlear field.

Reply: It is true that the expression of TRPA1 channels revealed by a knock-in reporter (PLAP) in the different cell types of the organ of Corti does not always follow the results of Ca²⁺-imaging or direct patch clamp experiments. However, we believe that it is important to provide PLAP data that show a possibility of the expression of TRPA1 channels in the different cell types. We have now emphasized throughout the manuscript that PLAP experiments do not show the actual expression of functional TRPA1 channels. Most importantly, we also emphasized that it does not provide any information on the intracellular sorting of TRPA1 channels between apical and basal sides of the sensory epithelium, which are separated by the tight junctions. That's why neither of our conclusions is based solely on PLAP labeling but rather follow the results of Ca²⁺-imaging and patch clamp experiments.

Specific comments.

Flufenamic acid potentiates Hensen's cells responses to 4-HNE, but the discussion around TRPA1 requiring synergistic stimulation for activation in Claudius cells is not coherent. A requirement for concerted actions by different agonists has not been suggested or demonstrated for TRPA1 before.

Reply: Following the Reviewer's comment, the speculation about concerted action of different TRPA1 agonists has been removed from the manuscript. In fact, it improved the manuscript significantly, since this speculation had nothing to do with the major conclusions of our study and, therefore, it has never been explored in detail.

*Flufenamic acid can stimulate TRPA1, but can also act on several other ion channels. Data demonstrating that the response evoked by 4-HNE in the presence of FFA can be inhibited by a TRPA1 antagonist would strengthen the conclusion from this experiment. Whereas there may be many possible explanations for why PLAP expression from the TRPA1 promoter in the *Trpa1*^{-/-} does not correlate precisely with functional TRPA1 in the *Trpa1*^{+/+}, the FFA results do not show that TRPA1 activation in the relevant cell types require "proper integration of several signals".*

Reply: As stated above, the relevant section has been removed from the manuscript.

A vast number of reactive molecules covalently modify and activate the channel. Many non-covalent ligands and divalent cations (in particular Ca²⁺ and Zn²⁺) also activate TRPA1 but what are the regulatory mechanisms the authors refer to in the discussion, 2nd paragraph, line 3?

Annexin A2 has been shown to limit the cellular pool of TRPA1 present in the membrane (the only negative modulator identified so far?). Are the authors suggesting a similar mechanism here?

Reply: We thank the Reviewer for their insightful thoughts, but we meant only the multiple mechanisms of TRPA1 activation rather than its true regulation. Therefore, we corrected our ambiguous sentence as following, “This is not surprising given that the TRPA1 channels could be activated through multiple signaling pathways by many non-covalent ligands and divalent cations, including its regulation by intracellular Ca²⁺...”

4-HNE is formed endogenously along with many other lipid peroxidation products and radicals under oxidative conditions. However, 4-HNE is not the most potent of TRPA1 agonists and in attempting to demonstrate the presence of functional TRPA1 channels, the authors may be better off using a more potent and more rapidly acting tool compound such as e.g. p-benzoquinone.

Reply. Following Reviewer’s comments, we have now included new data in the Supplementary Figure 2 showing the responses of Hensen’s cells to different concentrations of p-benzoquinone (pBQN).

TRPA1 antibodies generated non-specific staining: Is this staining non-specific or could it be an indication that some non-functional protein is made in the Trpa1^{-/-} mice?

Reply: We do not know. Our mouse strain carries genomic deletion of exons 21–25 of *Trpa1* that encode the S5 and S6 transmembrane domains of TRPA1 (Kwan et al, Neuron, 2006). Therefore, it is feasible that a truncated non-functional TRPA1 protein is still generated in these mice. However, in our hands, even the antibodies presumably targeting the C-terminus of TRPA1 produced some staining in the cochlea of *Trpa1^{-/-}* mice. That’s why we think that this staining was likely to be non-specific.

Discussion, first paragraph; the auditory nerve doesn't transmit nociceptive information, since auditory information is not normally transmitted to higher centres relevant for pain. The identity of the transduction molecule responsible for action potential firing doesn't define what sensory modality is transmitted.

Reply: We have removed this sentence.

The authors observe marked uptake of FM1-43 in OHC, but no Ca²⁺-responses evoked by TRPA1 agonists in the same cells. In contrast, Hensen's cells, which do display TRPA1 mediated Ca²⁺-influx responses, show very little evidence of FM1-43 uptake. These results do not support FM1-43 entry through TRPA1. These results rather seem to suggest that FM1-43 has labelled vesicles in OHC (the conventional use for FM1-43 and related styryl dyes), rather than entered the cells via TRPA1.

Reply: After repeating these experiments, we believe now that the Reviewer was right, and our FM1-43 data were inconclusive. Therefore, we have removed them from the manuscript.

Results Page 6, 2nd paragraph: The temporal pattern of the Ca²⁺-responses evoked by 4-HNE in the Kolliker's organ is very different from that observed in Hensen's cells. This is not sufficient to conclude that they are not mediated by TRPA1 activation. Near the bottom of page 8, the explanation for this response in Kolliker's organ is provided.

Reply: This misleading conclusion has been removed from the manuscript.

Reviewer #2 (Remarks to the Author):

The manuscript is interesting but also has significant drawbacks. The authors conclude that they have identified a new mechanism that regulates cochlear sensitivity that depends on TrpA1. They conclude that endogenous noise-induced compounds activate TrpA1, which leads to shape changes in the cochlea and inhibits cochlear amplification. They argue that TrpA1 in Hensen's cells have a central function in this process. These are farreaching conclusion that are not supported by direct experimental evidence.

Reply: We have performed number of additional *in vivo* experiments. Our updated results (Fig. 8 and relevant supplements) show that the absence of TRPA1 channels causes larger but less prolonged noise-induced temporary shift of hearing thresholds (TTS) *i.e.*, TRPA1 channels contribute to the regulation of “cochlear sensitivity” after acoustic trauma. This regulation seems to be protective, because without TRPA1 channels, even very mild acoustic trauma generates permanent changes in the latency of ABR wave I responses without any changes in the number of IHC synapses. Furthermore, we show that “noise-induced compounds” such as 4-HNE do activate TRPA1 channels in the supporting cells, leading to shape changes in the cochlea. By obvious technical reasons, the latter data were obtained only in the young postnatal cochlear explants. Therefore, it is still possible that the observed phenomena are different in the adult animals. In this revised version of the manuscript, we show also new data demonstrating the direct activation of Hensen’s cells by TRPA1 agonists in the adult cochlea (Fig 2g-i).

From the data in the manuscript it is not entirely clear which cells express the functionally relevant TrpA1 receptor. Gene expression studies (PLAP knock-in) do not seem to agree with the results obtained by identifying cells expressing TrpA1 using pharmacological activators. Several cell types seem to respond to TrpA1 agonists and while the signal in Hensen's cells is particularly strong, other cell types also show a significant response.

Reply: Certainly, PLAP reporter data do not always follow our functional assays. This is not surprising though since gene expression level does not always coincide with the amount of protein expression. Furthermore, gene expression cannot tell anything about protein trafficking to the apical vs. basal part of the cell within the sensory epithelium and about assembling a functional ion channel. That’s why neither of our conclusions is based solely on PLAP labeling but rather follow the results of more functionally relevant Ca^{2+} -imaging and patch clamp experiments. The latter experiments clearly show activation of Hensen’s cells first and then the secondary propagation of Ca^{2+} signals to other supporting cells, at least when TRPA1 agonists are applied from endolymphatic side of the sensory epithelium. This provides at least one likely cellular mechanism of TRPA1 signaling within the cochlea. We never claim that this is the only possible mechanism, since we do not have a reliable way of applying TRPA1 agonists to the perilymphatic side without disrupting the tissue, and therefore the actual cellular pattern of TRPA1 responses *in vivo* may be more complex. This is now clearly stated in the revised version of the Discussion.

While pharmacological activation of TrpA1 leads to shape changes in inner ear organ cultures, the extent to which this is relevant to the observed protective effect to noise in vivo remains speculative. It is not clear whether noise induces TrpA1 function in vivo and causes shape changes within the organ of Corti. There is also no direct link between the shape changes and protection. Thus the results are correlative and do not show causality.

Reply: We are puzzled by this comment. We show the changes of noise-induced TTS and the changes of TRPA1-initiated cochlear shape changes in the mice lacking TRPA1 channels. This is the best causality test for the role of TRPA1 channels, especially having in mind that *Trpa1*^{-/-} mice have perfectly normal hearing thresholds in control and, hence, no abnormalities of any essential cochlear structures. Perhaps, the Reviewer meant that our manuscript does not demonstrate cochlear shape changes after noise exposure? Yet, the long-lasting noise-induced contraction of the organ of Corti in the region encompassing Hensen’s and Deiters’ cells was demonstrated *in situ* in the isolated cochlea preparations (Flock et al., *J. Neurosci.*, 1999; Fridberger et al., *Biophys. J.*, 2004). Furthermore, Liberman’s group

reported a similar organ of Corti contraction *in vivo* in the live animals, 24h after mild noise that produces only TTS (Wang et al., *JARO*, 2002). The underlying mechanisms of this phenomenon are yet unknown, but our data suggest that TRPA1-mediated signaling may contribute to it. Furthermore, recent data show that the direct optogenetic activation of the Deiters' and outer pillar cells is protective (Lukashkina et al., *J. Neurosci.*, 2022). Hence, it is not surprising that activation of the same cells through propagation of TRPA1 signaling may be also protective. We have revised the Discussion to include all the above considerations.

The link to cochlear amplification is far fetched and not supported by experimental evidence. As such, the manuscript neither clearly defines the function of TrpA1 in the inner ear nor identifies a new mechanism that regulates cochlear sensitivity. Thus, while the findings are certainly worthwhile reporting, they seem to be suitable for a more specialized journal.

Reply: We respectfully disagree. As of today, the mechanisms of noise-induced TTS are largely speculative. Of course, our manuscript cannot clarify all of them. However, the manuscript provides clear evidence for the involvement of TRPA1 signaling in TTS and provides potential cellular mechanisms of how TRPA1 channels may do it. We believe this is an important contribution not only for the auditory research but also for sensory biology in general, having in mind the interest to noise-induced hearing loss in our society.

Reviewer #3 (Remarks to the Author):

In this paper the authors provide evidence that products of oxidative stress act on TRPA1 channels to alter cochlear mechanics and sensitivity over the course of several days. The majority of experiments are convincing and the significance of the findings high. Temporary loss of sensitivity (TTS - temporary threshold shift) is a common consequence of acoustic trauma and this work provides novel insights into underlying molecular mechanisms. The authors are to be commended for the extensive documentation they provide - the main body of the paper is elegantly illustrated and clearly described. However, a general problem is that the eight supplemental figures are provided with far less depth of description and rigor than the eight figures that make up the main document. For example, no statistical treatment is given for results reported in Supplemental Figures. At the same time, these provide anecdotal support for a number of interesting findings. If properly described and quantified these could constitute another interesting and strong publication in its own right. Perhaps the temptation was to add as much support as possible to the main conclusions. However, there is a cost in the struggle imposed on the Reader to weave together a fragmented presentation with varying levels of proof.

Reply: We have updated supplementary figures and included statistical analysis whenever it is possible. For the remaining supplementary figures, statistical analysis is provided either in the main relevant figure (Supplementary Figures 1 and 5) or in text (Supplementary Figures 3 and 4).

Some trimming and/or editing also would help the supplementary videos. Video #1 for example is unnecessary since video #2 incorporates the same point (and is by far the most impressive and useful of the supplementary materials). Video #3 would be much improved with labels to help the viewer identify cell types. Video #4 was quite confusing. In addition to also needing labels, it appears that movement precedes, as well as follows, the appearance of the white arrow. Which is the significant bit?

Reply: Although visually similar, videos 1 and 2 serve different purposes. Video 1 shows activation of Hensen's cells, while video 2 illustrates the propagation of the signal from Hensen's cells to Kolliker's organ. It would be confusing to reference video 2 during description of Hensen's cell responses (in connection with the main Figure 2). We have added the labels in the video 3, as requested. We have also labelled video 4 as requested. Video 4 was chosen because it shows not only TRPA1-evoked contraction on a "clean" background (as video 3 does) but also the presence of spontaneous background movements. To clarify, we have added the following sentence to the legend to the video: "Note that the TRPA1-

evoked contraction occurs at the background of the smaller spontaneous movements (for quantification of the relative amplitudes of these effects, see Figure 6c)".

TrpA1 null mice have greater initial loss of sensitivity, but more rapid recovery after acoustic trauma. From this latter effect the authors draw the somewhat surprising conclusion that longer-term TrpA1-mediated changes are protective to the cochlea. Isn't temporary threshold shift a sign of cochlear damage? If longer-lasting in TRPA1 wild-types then one ordinarily would conclude there was more damage???

Reply: Not necessarily. In fact, there are number of reasons to believe that temporary threshold shift represents a combination of the recoverable damage with the endogenous protective reactions within the cochlea. Our study argues that TRPA1-initiated responses may contribute to these protective reactions. We have now clarified it better in Discussion.

A final general point is that all their ex vivo studies examine cellular changes over the course of seconds to minutes. However from this they extrapolate to a days-long effect after acoustic trauma. They suggest that production of TrpA1 agonists continues for days, and show immunolabeling experiments in support of that proposal. This last supplemental figure is a good example of the general problem mentioned above. There is no critical analysis of these results, no mention of number of biological repetitions, and scant description of the methods or controls employed (e.g., "The specificity of the anti-HNE antibodies was tested in control experiments where we exposed postnatal cochlear explants to increasing concentrations (0, 0.02, 0.2 and 2 mM) of H2O2 (Invitrogen) for 4 to 6 h."). What else might the antibody bind to in treated tissue?

Reply: We have been also puzzled by the reported data on the delayed accumulation of endogenous TRPA1 agonists throughout days after a single incident of the acoustic trauma. That's why we have repeated these published experiments by ourselves, even though it was beyond the scope of our investigation. We still do not understand what exactly is driving such delayed accumulation of the byproducts of oxidative stress. However, independent of their origin, the 4-HNE byproduct would activate TRPA1 channels *in vivo* for a long time after noise exposure. The legend to the supplementary figure 6 now specifies that 4-HNE immunolabeling was investigated in three independent series of mice subjected to noise trauma. As to the specificity of labeling is concerned, we have used well-characterized antibodies (EMD Millipore), tested them in the cochlear explants exposed to oxidative stress with increasing concentrations of H₂O₂, and recapitulated previously published data (Yamashita et al., 2004).

Abstract:

"...novel mechanism for regulating hearing sensitivity after acoustic trauma..." why not a novel mechanism for temporary threshold shift?

Reply: Per Journal instructions, we aim our abstract to broad audience and, therefore, we prefer to avoid a field-specific term "temporary threshold shift". In addition, as explained before, TTS is likely to be a combination of recoverable cellular damage with endogenous protective mechanisms, TRPA1-mediated effects contribute to the latter mechanisms and, therefore, represent only a part of TTS.

Introduction:

*"Although the inner ear is believed to have no nociceptors..." but see recent publications such as Flores EN, Duggan A, Madathany T, Hogan AK, Márquez FG, Kumar G, Seal RP, Edwards RH, Liberman MC, García-Añoveros J. A non-canonical pathway from cochlea to brain signals tissue-damaging noise. *Curr Biol.* 2015 Mar 2;25(5):606-12. PMID: PMC4348215.*

Reply: This controversial statement has been removed.

"TRPA1 null mice exhibit normal hearing" - so is proposed protective effect of TRPA1 not evident?

Reply: As explained above, TRPA1 is likely to contribute to the endogenous regulatory component of TTS. This contribution cannot be detected until mice are exposed to loud noise.

Results:

PLAP under TRPA1 promoter - what was positive control? What was negative control? Would be helpful to have some independent evidence that the promoter construct actually drove specific expression - if possible.

Reply: We may only assume that PLAP expression indicate potential expression of TRPA1. That's why we performed more functional Ca²⁺-imaging and patch clamp assays to draw any conclusions in this study.

Rise of intracellular calcium very slow. Can this be due to influx across the plasma membrane? If so, how can it take such a long time (38 seconds)?

Reply: Slow opening seems to be a characteristic feature of TRPA1 channels. Even in heterologous systems, the first signs of TRPA1-activated current are initiated quite fast, but the current develops to the maximum within tens of seconds (Figure 4b, inset).

"Despite the high frequency of spontaneous Ca waves in the Kolliker's organ..." How about different wording here? "Although spontaneous calcium waves occurred frequently in Kolliker's organ, nonetheless it was possible to correlate 4-HNE exposure with the triggering of calcium waves.

Reply: We have changed the wording to the following: "Although spontaneous Ca²⁺ waves in the epithelium generated short-lived tissue contractions and complicated the analysis, on average, there was a noticeable increase in tissue movements during and after the TRPA1 agonist application..."

"...release of ATP into the extracellular medium..." check references for this statement - are these the correct ones?

Reply: Wrong reference has been replaced to Munoz et al., 2001.

Heterologous expression in HEK cells. While this is an interesting experiment in its own right, it is not conclusive (no proof that ATP acts through the TRPA1, simply correlated with expression, and no test of signaling pathway), and not proof of anything in cochlear cells. This figure could be left out.

Reply: We believe it is better to show these data. Without them, it is harder to explain why TRPA1 deficiency results in a larger shift of hearing thresholds during and immediately after noise exposure. On the other hand, it has been well established that TRPA1 channels could be activated downstream of various G-protein coupled receptors (Bandell et al., 2004; Jordt et al., 2004). Therefore, spending additional effort on confirming a similar signaling pathway downstream of P2Y receptors is not well justified.

"...no obvious alterations in whole-cell current...which argues against cell volume changes due to ion fluxes..." What about loss of KCl (charge neutral)? One or the other ion specifically-gated, the other follows passively. The time course of change is certainly slow enough.

Reply: This confusing statement has been removed.

Discussion:

"TRPA1 channels do not generate specific nociceptive responses in auditory nerve fibres." How do you know this? Were recordings made showing no increased activity in type I or type II afferents?

Reply: We are very thankful to the Reviewer for this comment! In fact, we have now obtained the data indicating that TRPA1 activation may produce also afferent signals in the type II afferents. This is a subject of a separate ongoing study. Obviously, we have removed the misleading statement from the manuscript.

"...causing physiologically significant tissue displacements..." Justified statement? How about just 'visible' displacements?

Reply: Corrected as suggested.

"...in older mice because it is likely to involve gap junctional conductance between inner and outer pillar cells that disappears in adult animals." Not to mention that Kolliker's organ disappears!

Reply: Corrected as following: "We do not believe that secondary activation of ATP-dependent Ca^{2+} waves in the Kolliker's organ (Fig.4a; Supplementary Fig.4a,b) is relevant to the adult animals, first, because Kolliker's organ disappear/remodel in adults and, second, because it is likely to involve the gap junctional conductance between inner and outer pillar cells that also disappears in adult animals..."

Figure 1. specify age illustrated in A. Specify z-axis illustrated in B -orthogonal to plane of previous image?

Reply: Done. Figure legend now indicates that this is a drawing of the cochlear epithelium in young postnatal mice (<P7).

Figure 6. Put labeled axis on panel 'c' to show where 7 minute integral was collected for panel 'd'. Otherwise the eye is drawn just to the time of drug application and the effect is not convincing.

Reply. We have changed this figure as requested (now Figure 6). X axis in the panel c now shows the 7 min time regions before and after drug application that were used to calculate the data on panel d.

Figure 6e. which diameter measured - at what point along cell?

Reply. We have added to the figure legend that the cell diameters were measured at the lower level of Deiters' cells.

Figure 7b. specify time after UV flash that image was taken.

Reply. The labels (1) and (2) on panel c now indicate the timing where the images from panel b were taken.

REVIEWERS' COMMENTS

Reviewer #1 (Remarks to the Author):

It was a pleasure to see a revised version of this manuscript. Since it has been some time since it was first submitted, I have on occasion, in vain, looked for the paper elsewhere. The authors have addressed all major comments on the first submission very effectively, and the paper has consequently been improved markedly. Apart from the very minor point below, I have no further comments or questions that I would like addressed.

Minor comment

Discussion, 2nd paragraph, ln 2: It is not clear that the fact that TRPA1 can be activated by several pathways, does not perhaps explain the differential responsiveness in different cells. On the other hand, differences in receptor or kinase modulation, accessory proteins (not all identified) and even differences in Ca²⁺-handling (as indicated in fig. 3b) etc could determine responsiveness?

Reviewer #3 (Remarks to the Author):

The authors have adequately answered my concerns. No further comments.

We thank the Reviewers for their thoughtful and constructive comments, and we are glad that we successfully addressed all the points raised in the previous round of review. The answer to the remaining Reviewer's comment is given below.

Reviewer #1 (Remarks to the Author):

It was a pleasure to see a revised version of this manuscript. Since it has been some time since it was first submitted, I have on occasion, in vain, looked for the paper elsewhere. The authors have addressed all major comments on the first submission very effectively, and the paper has consequently been improved markedly. Apart from the very minor point below, I have no further comments or questions that I would like addressed.

Minor comment

Discussion, 2nd paragraph, ln 2: It is not clear that the fact that TRPA1 can be activated by several pathways, does not perhaps explain the differential responsiveness in different cells. On the other hand, differences in receptor or kinase modulation, accessory proteins (not all identified) and even differences in Ca²⁺-handling (as indicated in fig. 3b) etc could determine responsiveness?

Following Reviewer's comment, we realized that the corresponding sentence in Discussion was indeed confusing, since it could make a false impression that differential responsiveness of the cells results mostly from different activation mechanisms. Therefore, we have modified it as following: "This is not surprising given that the TRPA1 channels could be activated by many non-covalent ligands and divalent cations and regulated through multiple signaling pathways (refs), including its regulation by intracellular Ca²⁺ (Fig.3)."

Reviewer #3 (Remarks to the Author):

The authors have adequately answered my concerns. No further comments.

We thank the Reviewer for the constructive critique in the previous round.